# Dual Gauss-Newton Directions for Deep Learning

## Abstract

Gauss-Newton (a.k.a. prox-linear) directions can be computed by solving an optimization subproblem that trade-offs between a partial linearization of the objective function and a proximity term. In this paper, we study the possibility to leverage the convexity of this subproblem in order to instead solve the corresponding dual. As we show, the dual can be advantageous when the number of network outputs is smaller than the number of network parameters. We propose a conjugate gradient algorithm to solve the dual, that integrates seamlessly with autodiff through the use of linear operators and handles dual constraints. We prove that this algorithm produces descent directions, when run for any number of steps. Finally, we study empirically the advantages and current limitations of our approach compared to various popular deep learning solvers.

## 1 Introduction

We consider deep learning objectives of the form

$$\min_{\boldsymbol{w} \in \mathbb{R}^p} \left[ \frac{1}{n} \sum_{i=1}^n h_i(\boldsymbol{w}) = \frac{1}{n} \sum_{i=1}^n \ell_i(f_i(\boldsymbol{w})) \right], \tag{1}$$

where $h_i \coloneqq \ell_i \circ f_i$, $\ell_i \colon \mathbb{R}^k \to \mathbb{R}$ is a convex loss on a sample $i \in [n]$, and $f_i \colon \mathbb{R}^p \to \mathbb{R}^k$ is a neural network with parameters $\boldsymbol{w} \in \mathbb{R}^p$, applied on the sample $i$. Such objectives are generally tackled by algorithms ranging from stochastic gradient descent to adaptive methods, incorporating momentum (Sutskever et al., 2013; Kingma & Ba, 2015) or linesearch (Vaswani et al., 2019). Stochastic gradients can be computed efficiently using autodiff frameworks such as JAX (Bradbury et al., 2018) or Pytorch (Paszke et al., 2019). However, the information they provide on the objective function is intrinsically limited, since they amount to using a **full linearization** of $h_i$.

In this paper, we consider constructing update directions through a **partial linearization** of the function $h_i$. Such an approach has a long history starting from the Gauss-Newton and Levenberg-Marquardt methods (Levenberg, 1944; Marquardt, 1963; Björck, 1996; Kelley, 1999), used for nonlinear least squares. The extension to arbitrary convex loss functions of this approach has been called **modified Gauss-Newton** (Burke, 1985; Nesterov, 2007; Tran-Dinh et al., 2020; Zhang & Xiao, 2021) or **prox-linear** (Drusvyatskiy & Paquette, 2019; Pillutla et al., 2019), name that we will use in the rest of this paper. While Drusvyatskiy & Paquette (2019) considered the dual formulation of prox-linear to derive convergence rates, Drusvyatskiy & Paquette (2019) did not develop an algorithm based on the dual formulation for stochastic optimization. In the context of deep learning, several authors have considered tackling Gauss-Newton-like oracles by decomposing the resulting problem through the layers of a deep network (Yu & Wilamowski, 2018; Martens & Grosse, 2015; Botev et al., 2017; Gupta et al., 2018; Anil et al., 2020), see Appendix A for a detailed literature review.

In this paper, we propose instead to exploit the **convexity** of the subproblem associated with such prox-linear directions by switching to their **dual formulation**. We make the following contributions.

- After reviewing prox-linear (a.k.a. modified Gauss-Newton) directions, computed by solving an optimization subproblem that trade-offs between a partial linearization of the objective function and a proximity term, we cleanly derive the subproblem's **Fenchel-Rockafellar dual**.

- To solve this dual, we present a **conjugate gradient** algorithm. Our proposal integrates seamlessly in an autodiff framework, by leveraging Jacobian-vector products (JVP) and vector-Jacobian products (VJP) and can handle dual constraints.
- We prove that the proposed algorithm produces a **descent direction**, when run for any number of steps, enabling its use as a drop-in replacement for the stochastic gradient in existing algorithms, such as SGD.
- Finally, we present comprehensive empirical results demonstrating the **advantages** and **current limitations** of our approach.

## 2 Prox-linear directions via the primal

In this section, we review prox-linear directions, which are based on the idea of **partial linearization**.

### 2.1 Variational perspective

To motivate prox-linear directions, we review the variational perspective on gradient and Newton directions. Let the **linear approximation** of $h_i$ around $\boldsymbol{w}^t$ be

$$\mathrm{lin}(h_i, \boldsymbol{w}^t)(\boldsymbol{w}) := h_i(\boldsymbol{w}^t) + \langle \nabla h_i(\boldsymbol{w}^t), \boldsymbol{w} - \boldsymbol{w}^t \rangle \approx h_i(\boldsymbol{w}).$$

From a variational perspective, the **stochastic gradient** can be seen as the minimization of this linear approximation and of a quadratic regularization term,

$$\gamma \nabla h_i(\boldsymbol{w}^t) = \operatorname*{argmin}_{\boldsymbol{d} \in \mathbb{R}^p} \mathrm{lin}(h_i, \boldsymbol{w}^t)(\boldsymbol{w}^t - \boldsymbol{d}) + \frac{1}{2\gamma} \|\boldsymbol{d}\|_2^2.$$

The information provided by a gradient is therefore limited by the quality of a linear approximation of the objective. Alternatively, we may consider the **quadratic approximation** of $h_i$ around $\boldsymbol{w}^t$,

$$\mathrm{quad}(h_i, \boldsymbol{w}^t)(\boldsymbol{w}) := h_i(\boldsymbol{w}^t) + \langle \nabla h_i(\boldsymbol{w}^t), \boldsymbol{w} - \boldsymbol{w}^t \rangle + \frac{1}{2} \langle \boldsymbol{w} - \boldsymbol{w}^t, \nabla^2 h_i(\boldsymbol{w}^t)(\boldsymbol{w} - \boldsymbol{w}^t) \rangle.$$

The **regularized Newton direction** is then

$$\operatorname*{argmin}_{\boldsymbol{d} \in \mathbb{R}^p} \left\{ \mathrm{quad}(h_i, \boldsymbol{w}^t)(\boldsymbol{w}^t - \boldsymbol{d}) + \frac{1}{2\gamma} \|\boldsymbol{d}\|_2^2 \right\} = (\nabla^2 h_i(\boldsymbol{w}^t) + \gamma^{-1} \mathrm{I})^{-1} \nabla h_i(\boldsymbol{w}^t).$$

Unfortunately, when $f_i$ is a neural network, $h_i = \ell_i \circ f_i$ is typically nonconvex and the Hessian $\nabla^2 h_i(\boldsymbol{w})$ is an **indefinite** matrix. Therefore, the minimization above is that of a **nonconvex quadratic** subproblem, that may be hard to solve. Furthermore, the obtained direction is **not** guaranteed to define a **descent direction**.

### 2.2 Convex-linear approximations

Instead of **fully linearizing** $h_i$, which amounts to linearizing **both** $\ell_i$ and $f_i$, we may linearize **only** $f_i$ but keep $\ell_i$ as is. That is, we may use the **partial linear approximation** of $h_i$,

$$\begin{aligned}
\mathrm{plin}(\ell_i, f_i, \boldsymbol{w}^t)(\boldsymbol{w}) &:= \ell_i(f_i(\boldsymbol{w}^t) + \partial f_i(\boldsymbol{w}^t)(\boldsymbol{w} - \boldsymbol{w}^t)) \\
&= \ell_i(\boldsymbol{f}_i^t + J_i^t(\boldsymbol{w} - \boldsymbol{w}^t)),
\end{aligned}$$

where we defined the shorthands

$$\boldsymbol{f}_i^t := f_i(\boldsymbol{w}^t) \quad \text{and} \quad J_i^t := \partial f_i(\boldsymbol{w}^t).$$

We say that the approximation is **convex-linear**, since it is the composition of the convex $\ell_i$ and of the linear approximation of $f_i$. We view $J_i^t$ as a **linear map** from $\mathbb{R}^p$ to $\mathbb{R}^k$, with **adjoint operator** $(J_i^t)^*$, a linear map from $\mathbb{R}^k$ to $\mathbb{R}^p$. That is, we assume that we can compute the Jacobian-vector product (JVP) $(J_i^t)\boldsymbol{u}$ for any direction $\boldsymbol{u} \in \mathbb{R}^p$ and the vector-Jacobian product (VJP) $(J_i^t)^*\boldsymbol{v}_i$ for any direction $\boldsymbol{v}_i \in \mathbb{R}^k$. This will be the case if $f_i$ is implemented using an autodiff framework such as JAX or PyTorch.

The minimization of the convex-linear approximation and of a quadratic regularization term leads to the definition of the **prox-linear** (a.k.a. **modified Gauss-Newton**) **direction**:

$$\boldsymbol{d}(\gamma\ell_i, f_i)(\boldsymbol{w}^t) := \underset{\boldsymbol{d}\in\mathbb{R}^p}{\operatorname{argmin}}\, \operatorname{plin}(\ell_i, f_i, \boldsymbol{w}^t)(\boldsymbol{w}^t - \boldsymbol{d}) + \frac{1}{2\gamma}\|\boldsymbol{d}\|_2^2$$

$$= \underset{\boldsymbol{d}\in\mathbb{R}^p}{\operatorname{argmin}}\, \ell_i(\boldsymbol{f}_i^t - J_i^t\boldsymbol{d}) + \frac{1}{2\gamma}\|\boldsymbol{d}\|_2^2. \tag{2}$$

Typically, we will only obtain an **approximate solution**, that we denote $\boldsymbol{d}_i^t \approx \boldsymbol{d}(\gamma\ell_i, f_i)(\boldsymbol{w}^t)$. Once we obtained $\boldsymbol{d}_i^t$, we can update the parameters as

$$\boldsymbol{w}^{t+1} := \boldsymbol{w}^t - \boldsymbol{d}_i^t.$$

We emphasize, however, that unlike the gradient, the prox-linear direction is not linear in $\gamma$, i.e.,

$$\boldsymbol{w}^t - \boldsymbol{d}(\gamma\ell_i, f_i)(\boldsymbol{w}^t) \neq \boldsymbol{w}^t - \gamma\boldsymbol{d}(\ell_i, f_i)(\boldsymbol{w}^t).$$

**Mini-batch extension.** The prox-linear direction presented earlier is obtained from a single sample $i \in [n]$. We now consider the extension to a mini-batch $S := \{i_1, \ldots, i_m\} \subseteq [n]$ of size $m = |S|$. Let us define

$$f_S(\boldsymbol{w}) := (f_{i_1}(\boldsymbol{w}), \ldots, f_{i_m}(\boldsymbol{w}))^\top \in \mathbb{R}^{m\times k}, \qquad \ell_S(\boldsymbol{f}_S) := \sum_{j=1}^m \ell_{i_j}(\boldsymbol{f}_i) \in \mathbb{R}.$$

We then define the mini-batch direction

$$\boldsymbol{d}(\gamma\ell_S, f_S)(\boldsymbol{w}^t) := \underset{\boldsymbol{d}\in\mathbb{R}^p}{\operatorname{argmin}}\, \frac{1}{m}\sum_{i\in S}\operatorname{plin}(\ell_i, f_i)(\boldsymbol{w}^t) + \frac{1}{2\gamma}\|\boldsymbol{d}\|_2^2$$

$$= \underset{\boldsymbol{d}\in\mathbb{R}^p}{\operatorname{argmin}}\, \sum_{i\in S}\ell_i(\boldsymbol{f}_i^t - J_i^t\boldsymbol{d}) + \frac{m}{2\gamma}\|\boldsymbol{d}\|_2^2. \tag{3}$$

Typically, we only compute an approximate solution, denoted $\boldsymbol{d}_S^t \approx \boldsymbol{d}(\gamma\ell_S, f_S)(\boldsymbol{w}^t)$. The above subproblem is again convex. It can be solved directly in the primal, or, as we propose in Section 3, by switching to the dual. Once we obtained $\boldsymbol{d}_S^t$, we may perform the update

$$\boldsymbol{w}^{t+1} := \boldsymbol{w}^t - \boldsymbol{d}_S^t,$$

or use the direction $\boldsymbol{d}_S^t$ in an existing stochastic algorithm. Unlike the mini-batch stochastic gradient, the mini-batch prox-linear direction is not the average of the individual stochastic directions, i.e.,

$$\boldsymbol{d}(\ell_S, f_S)(\boldsymbol{w}^t) \neq \frac{1}{m}\sum_{i\in S}\boldsymbol{d}(\ell_i, f_i)(\boldsymbol{w}^t).$$

The mini-batch prox-linear direction can therefore take advantage of the correlations between the samples of the mini-batch, unlike the gradient oracle.

### 2.3 Quadratic-linear approximations

In the previous section, we studied convex-linear approximations, i.e., the composition of a convex $\ell_S$ and of the linear approximation of $f_S$, on a sample $S$. As in the so-called **"generalized" Gauss-Newton** algorithm (Gargiani et al., 2020), we can replace $\ell_S$ with its quadratic approximation around $f_S(\boldsymbol{w}^t)$, namely,

$$q_S^t := \operatorname{quad}(\ell_S, f_S(\boldsymbol{w}^t)).$$

Replacing $\ell_S$ with $q_S^t$, we get after simple algebraic manipulations (see Appendix C) a **convex quadratic**

$$\boldsymbol{d}(\gamma q_S^t, f_S)(\boldsymbol{w}^t) = \underset{\boldsymbol{d}\in\mathbb{R}^p}{\operatorname{argmin}}\, \frac{1}{m}\sum_{i\in S}q_i^t(\boldsymbol{f}_i^t - J_i^t\boldsymbol{d}) + \frac{1}{2\gamma}\|\boldsymbol{d}\|_2^2 \tag{4}$$

$$= \underset{\boldsymbol{d}\in\mathbb{R}^p}{\operatorname{argmin}}\, \frac{1}{2}\langle\boldsymbol{d}, (J_S^t)^* H_S^t J_S^t\boldsymbol{d}\rangle - \langle(J_S^t)^*\boldsymbol{g}_S^t, \boldsymbol{d}\rangle + \frac{m}{2\gamma}\|\boldsymbol{d}\|_2^2$$

$$\approx \boldsymbol{d}(\gamma\ell_S, f_S)(\boldsymbol{w}^t),$$

where, denoting $\boldsymbol{f}_S^t := f_S(\boldsymbol{w}^t)$, we defined

$$J_S^t \boldsymbol{u} := \partial \boldsymbol{f}_S(\boldsymbol{w}^t)\boldsymbol{u} = (J_{i_1}^t \boldsymbol{u}, \ldots, J_{i_m}^t \boldsymbol{u})^\top$$

$$(J_S^t)^* \boldsymbol{v} := \partial \boldsymbol{f}_S(\boldsymbol{w}^t)^* \boldsymbol{v} = \sum_{j=1}^m (J_{i_j}^t)^* \boldsymbol{v}_j$$

$$H_S^t \boldsymbol{v} := \nabla \ell_S^2(\boldsymbol{f}_S^t)\boldsymbol{v} = \left( \nabla^2 \ell_{i_1}(\boldsymbol{f}_1^t)\boldsymbol{v}_1, \ldots, \nabla^2 \ell_{i_m}(\boldsymbol{f}_m^t)\boldsymbol{v}_m \right)$$

$$\boldsymbol{g}_S^t := \nabla \ell_S(\boldsymbol{f}_S^t) = (\nabla \ell_{i_1}(\boldsymbol{f}_1^t), \ldots, \nabla \ell_{i_m}(\boldsymbol{f}_m^t)).$$

In contrast, we recall that the subproblem associated with the Newton direction is a **nonconvex quadratic**.

Without a regularization term, that is by using $\gamma = +\infty$ in (4), $\boldsymbol{d}(q_S^t, f_S)$ amounts to a generalized Gauss-Newton step (Gargiani et al., 2020), which itself matches a natural gradient step (Amari, 1998) if $\ell_i$ is the negative log-likelihood of a regular exponential family (Martens, 2020).

**Examples.** For the **squared loss**, we have

$$\ell_i(\boldsymbol{f}_i) := \frac{1}{2}\|\boldsymbol{f}_i - \boldsymbol{y}_i\|_2^2, \ \nabla \ell_i(\boldsymbol{f}_i) = \boldsymbol{f}_i - \boldsymbol{y}_i,$$

$$\nabla^2 \ell_i(\boldsymbol{f}_i) = \mathrm{I}, \qquad \nabla^2 \ell_i(\boldsymbol{f}_i)\boldsymbol{v}_i = \boldsymbol{v}_i.$$

For the **logistic loss**, we have

$$\ell_i(\boldsymbol{f}_i) := \mathrm{LSE}(\boldsymbol{f}_i) - \langle \boldsymbol{f}_i, \boldsymbol{y}_i \rangle, \ \nabla \ell_i(\boldsymbol{f}_i) = \sigma(\boldsymbol{f}_i) - \boldsymbol{y}_i,$$

$$\nabla^2 \ell_i(\boldsymbol{f}_i) = \mathbf{diag}(\sigma(\boldsymbol{f}_i)) - \sigma(\boldsymbol{f}_i)\sigma(\boldsymbol{f}_i)^\top,$$

$$\nabla^2 \ell_i(\boldsymbol{f}_i)\boldsymbol{v}_i = \sigma(\boldsymbol{f}_i) \odot \boldsymbol{v}_i - \langle \boldsymbol{v}_i, \sigma(\boldsymbol{f}_i) \rangle \sigma(\boldsymbol{f}_i),$$

where $\sigma(\boldsymbol{f}_i) := \mathrm{softmax}(\boldsymbol{f}_i)$.

### 2.4 Practical implementation

**Computing an approximate direction.** As we saw, computing a direction involves the (approximate) resolution of the convex subproblem (3) or of the convex quadratic subproblem (4). In general, we must resort to an iterative convex optimization algorithm.

On one extreme, performing a **single** gradient step on the convex subproblem would bring no advantage, since it would be equivalent to a gradient step on the nonconvex original problem. Indeed, we have

$$\nabla[\mathrm{plin}(\ell_i, f_i, \boldsymbol{w}^t)](\boldsymbol{w}^t) = \nabla h_i(\boldsymbol{w}^t) = (J_i^t)^* \nabla \ell_i(f_i(\boldsymbol{w}^t)),$$

and generally $\nabla[\mathrm{plin}(\ell_S, f_S, \boldsymbol{w}^t)](\boldsymbol{w}^t) = \nabla h_S(\boldsymbol{w}^t)$.

On the other extreme, if we solve the subproblem to high accuracy, the overhead of solving the subproblem may hinder the benefit of a better direction than the gradient. A trade-off must be struck between the additional computational complexity of the subproblem and the benefits of a refined direction (Lin et al., 2018; Drusvyatskiy & Paquette, 2019).

We argue that a good inner solver should meet the following requirements.

1. It should be easy to implement or widely available.

2. It should be compatible with linear maps, i.e., it should not require materializing $J_S^t$ as a matrix in memory. Such an algorithm is called **matrix-free**.

3. It should not require tuning hyperparameters.

4. It should leverage the specificities of the subproblem.

If we decide to solve the primal instead, as was done in the existing literature, LBFGS (Nocedal & Wright, 1999) is a good generic candidate, but that does not especially leverage the nature of the subproblem. In

the full-batch setting, where $m = n$, Drusvyatskiy & Paquette (2019); Pillutla et al. (2023) considered variance-reduced algorithms such as SVRG. In the case of the convex quadratic (4), we can use the conjugate gradient method.

Whatever the inner algorithm used, we can control the trade-off between computational cost and precision using a maximum number of inner iterations.

**Performing an update.** Once we obtained an approximate direction $\boldsymbol{d}_S^t \approx \boldsymbol{d}(\gamma \ell_S, f_S)(\boldsymbol{w}^t)$, we already saw that we can simply perform the update

$$\boldsymbol{w}^{t+1} := \boldsymbol{w}^t - \boldsymbol{d}_S^t. \tag{5}$$

This may require the tuning of the regularization parameter $\gamma$, which effectively act as a stepsize by analogy with the variational formulation of the gradient.

More generally, we can use $\boldsymbol{d}_S^t$ as a **drop-in replacement** for the stochastic gradient direction in other algorithms such as SGD with momentum or SGD with linesearch. For example, we may fix the regularization parameter to some value, say $\gamma = 1$, and instead perform the update

$$\boldsymbol{w}^{t+1} := \boldsymbol{w}^t - \eta^t \boldsymbol{d}_S^t, \tag{6}$$

where $\eta^t$ is a stepsize (we use a different letter to distinguish it from the regularization parameter $\gamma$), selected by a backtracking Armijo linesearch. That is, we seek $\eta^t$ satisfying

$$h_S(\boldsymbol{w}^t - \eta^t \boldsymbol{d}_S^t) \leq h_S(\boldsymbol{w}^t) - \beta \eta^t \langle \boldsymbol{d}_S^t, \boldsymbol{g}_S^t \rangle,$$

where $h_S = \frac{1}{m} \ell_S \circ f_S(\boldsymbol{w})$, $\boldsymbol{g}_S^t := \nabla h_S(\boldsymbol{w}^t) = \frac{1}{m} \sum_{i \in S} \nabla h_i(\boldsymbol{w}^t)$ is the mini-batch stochastic gradient evaluated at $\boldsymbol{w}^t$, and where $\beta \in (0, 1)$ is a standard constant, typically set to $\beta = 10^{-4}$. The entire procedure is summarized in Algorithm 1.

## 3 Prox-linear directions via the dual

In this section, we study how to obtain an approximate prox-linear direction, by solving the convex subproblem (3) or the convex quadratic subproblem (4) in the dual.

### 3.1 Convex-linear approximations

By taking advantage of the availability of the conjugate

$$\ell_i^*(\boldsymbol{\alpha}_i) := \sup_{\boldsymbol{f}_i \in \mathbb{R}^k} \langle \boldsymbol{f}_i, \boldsymbol{\alpha}_i \rangle - \ell_i(\boldsymbol{f}_i),$$

we can express the prox-linear direction (2) on a single sample $i \in [n]$ as

$$\boldsymbol{d}(\gamma \ell_i, f_i)(\boldsymbol{w}^t) = \gamma (J_i^t)^* \boldsymbol{\alpha}(\gamma \ell_i, f_i)(\boldsymbol{w}^t),$$

where we defined the solution of the dual of (2),

$$\boldsymbol{\alpha}(\gamma \ell_i, f_i)(\boldsymbol{w}^t) := \operatorname*{argmin}_{\boldsymbol{\alpha}_i \in \mathbb{R}^k} \ell_i^*(\boldsymbol{\alpha}_i) - \langle \boldsymbol{\alpha}_i, \boldsymbol{f}_i^t \rangle + \frac{\gamma}{2} \|(J_i^t)^* \boldsymbol{\alpha}_i\|_2^2.$$

Let us compare this with a stochastic gradient:

$$\gamma \nabla h_i(\boldsymbol{w}^t) = \gamma (J_i^t)^* \nabla \ell_i(\boldsymbol{f}_i^t).$$

The dual viewpoint reveals that the prox-linear direction can be seen as replacing the gradient of the loss $\nabla \ell_i(\boldsymbol{f}_i^t)$ by the solution of the subproblem's dual $\boldsymbol{\alpha}(\gamma \ell_i, f_i)(\boldsymbol{w}^t)$. This also suggests that $\nabla \ell_i(\boldsymbol{f}_i^t)$ is a **good initialization** for $\boldsymbol{\alpha}_i$.

---

**Algorithm 1** Primal-based prox-linear direction

---

1: **Inputs:** parameters $\boldsymbol{w}^t$, batch $S = \{i_1, \ldots, i_m\} \subseteq [n]$, regularization $\gamma > 0$ (set to 1 if linesearch is used)
2: Compute $\boldsymbol{f}_S^t = (f_{i_1}(\boldsymbol{w}^t), \ldots, f_{i_m}(\boldsymbol{w}^t))^\top \in \mathbb{R}^{m \times k}$
3: Instantiate JVP and VJP operators by autodiff:

$$\boldsymbol{u} \mapsto J_S^t \boldsymbol{u} = (J_{i_1}^t \boldsymbol{u}, \ldots, J_{i_m}^t \boldsymbol{u}) \in \mathbb{R}^{m \times k}, \quad \forall \boldsymbol{u} \in \mathbb{R}^p$$

$$\boldsymbol{v} \mapsto (J_S^t)^* \boldsymbol{v} = \sum_{j=1}^m (J_{i_j}^t)^* \boldsymbol{v}_j \in \mathbb{R}^p, \quad \forall \boldsymbol{v} \in \mathbb{R}^{m \times k}$$

4: Run inner solver to approximately solve (3), i.e.,

$$\boldsymbol{d}_S^t \approx \operatorname*{argmin}_{\boldsymbol{d} \in \mathbb{R}^p} \frac{1}{m} \sum_{i \in S} \ell_i(\boldsymbol{f}_i^t - J_i^t \boldsymbol{d}) + \frac{1}{2\gamma} \|\boldsymbol{d}\|_2^2,$$

or its quadratic approximation (4), i.e.,

$$\boldsymbol{d}_S^t \approx \operatorname*{argmin}_{\boldsymbol{d} \in \mathbb{R}^p} \frac{1}{m} \sum_{i \in S} q_i^t(\boldsymbol{f}_i^t - J_i^t \boldsymbol{d}) + \frac{1}{2\gamma} \|\boldsymbol{d}\|_2^2$$

5: Set next parameters $\boldsymbol{w}^{t+1}$ by

$$\boldsymbol{w}^{t+1} := \boldsymbol{w}^t - \boldsymbol{d}_S^t \qquad \text{(fixed stepsize (5))}$$
$$\text{or} \quad \boldsymbol{w}^{t+1} := \boldsymbol{w}^t - \eta^t \boldsymbol{d}_S^t \qquad \text{(linesearch (6))}$$

for $\eta^t$ s.t. $h_S(\boldsymbol{w}^t - \eta^t \boldsymbol{d}_S^t) \leq h_S(\boldsymbol{w}^t) - \beta \eta^t \langle \boldsymbol{d}_S^t, \boldsymbol{g}_S^t \rangle$.
6: **Output:** $\boldsymbol{w}^{t+1} \in \mathbb{R}^p$

---

**Benefit of using the dual.** The dual subproblem involves $k$ variables while the primal subproblem involves $p$ variables. Typically, $k$ is the number of network outputs (e.g., classes), while $p$ is the number of network parameters. The dual subproblem is therefore advantageous when $k \ll p$, which is often the case.

**Mini-batch extension.** For the mini-batch case, the prox-linear direction can be computed as

$$\boldsymbol{d}(\gamma \ell_S, f_S)(\boldsymbol{w}^t) = \frac{\gamma}{m}(J_S^t)^* \boldsymbol{\alpha}(\gamma \ell_S, f_S)(\boldsymbol{w}^t),$$

where we defined the dual solution of (3) by

$$\boldsymbol{\alpha}(\gamma \ell_S, f_S)(\boldsymbol{w}^t) := \operatorname*{argmin}_{\boldsymbol{\alpha}_S \in \mathbb{R}^{m \times k}} \ell_S^*(\boldsymbol{\alpha}_S) - \langle \boldsymbol{\alpha}_S, \boldsymbol{f}_S^t \rangle + \frac{\gamma}{2m} \|(J_S^t)^* \boldsymbol{\alpha}_S\|_2^2, \tag{7}$$

where $\ell_S^*(\boldsymbol{\alpha}_S) := \sum_{i \in S} \ell_i^*(\boldsymbol{\alpha}_i)$.

This time, the dual subproblem involves $m \times k$ variables, where $m$ is the mini-batch size and $k$ is the number of network outputs, while the primal subproblem involves $p$ variables, as before. If the mini-batch is not too large, we typically have $mk \ll p$ and therefore the dual subproblem is still advantageous. Algorithm 2 summarizes our approach.

**Examples.** For the **squared loss**, the conjugate is

$$\ell_i^*(\boldsymbol{\alpha}_i) = \frac{1}{2} \|\boldsymbol{\alpha}_i\|_2^2 + \langle \boldsymbol{\alpha}_i, \boldsymbol{y}_i \rangle.$$

The dual subproblem therefore becomes

$$\boldsymbol{\alpha}(\gamma \ell_S, f_S)(\boldsymbol{w}^t) = \operatorname*{argmin}_{\boldsymbol{\alpha}_S \in \mathbb{R}^{m \times k}} \sum_{i=1}^m \frac{1}{2} \|\boldsymbol{\alpha}_i\|_2^2 - \langle \boldsymbol{\alpha}_i, \boldsymbol{g}_i^t \rangle + \frac{\gamma}{2m} \|(J_S^t)^* \boldsymbol{\alpha}_S\|_2^2,$$

where $\boldsymbol{g}_i^t := \boldsymbol{f}_i^t - \boldsymbol{y}_i = \nabla \ell_i(\boldsymbol{f}_i^t)$. Setting the gradient to zero, this gives the linear system

$$\left(\frac{\gamma}{m} J_S^t (J_S^t)^* + \mathrm{I}\right) \boldsymbol{\alpha}_S^t = \boldsymbol{g}_S^t,$$

where $\boldsymbol{g}_S^t := (\boldsymbol{g}_i^t)_{i \in S} \in \mathbb{R}^{m \times k}$ and $\boldsymbol{\alpha}_S^t = \boldsymbol{\alpha}(\gamma \ell_S, f_S)(\boldsymbol{w}^t)$. We can solve this system using the conjugate gradient method. Note that if $\boldsymbol{\alpha}_i^t$ is equal to the residual $\boldsymbol{y}_i - \boldsymbol{f}_i^t$, then $(J_i^t)^* \boldsymbol{\alpha}_i^t = \nabla h_i(\boldsymbol{w}^t)$. Therefore, the residual is a good initialization for the conjugate gradient method.

For the **logistic loss**, the conjugate is

$$\ell_i^*(\boldsymbol{\alpha}_i) = \langle \boldsymbol{\mu}_i, \log \boldsymbol{\mu}_i \rangle + \iota_{\triangle^k}(\boldsymbol{\mu}_i), \quad \text{for } \boldsymbol{\mu}_i := \boldsymbol{y}_i + \boldsymbol{\alpha}_i,$$

where $\iota_{\triangle^k}$ is the indicator function of the probability simplex $\triangle^k := \{\boldsymbol{\mu} \in \mathbb{R}^k : \boldsymbol{\mu} \geq 0, \boldsymbol{\mu}^\top \mathbf{1} = \mathbf{1}\}$.

Applying this conjugate, we arrive at the subproblem

$$\boldsymbol{\mu}_S^t := \operatorname*{argmin}_{\substack{\boldsymbol{\mu}_S \in \mathbb{R}^{m \times k} \\ \boldsymbol{\mu}_i \in \triangle^k}} \langle \boldsymbol{\mu}_S, \log \boldsymbol{\mu}_S \rangle - \langle \boldsymbol{\mu}_S - \boldsymbol{y}_S, \boldsymbol{f}_S^t \rangle + \frac{\gamma}{2m} \|(J_S^t)^*(\boldsymbol{\mu}_S - \boldsymbol{y}_S)\|_2^2,$$

where $\boldsymbol{y}_S := (\boldsymbol{y}_{i_1}, \dots, \boldsymbol{y}_{i_m})^\top \in \mathbb{R}^{m \times k}$. This is a **constrained** convex optimization problem, that can be solved by, e.g., projected gradient descent. Changing the variable back, we obtain

$$\boldsymbol{\alpha}(\gamma \ell_S, f_S)(\boldsymbol{w}^t) = \boldsymbol{\mu}_S^t - \boldsymbol{y}_S.$$

## 3.2 Quadratic-linear approximations

To enable the use of the conjugate gradient method, we consider the dual of the quadratic approximation presented earlier. If the Hessian of the loss $\ell_i$ is invertible, the dual solution $\boldsymbol{\alpha}(\gamma q_S^t, f_S)(\boldsymbol{w}^t)$ of (4) equals

$$\operatorname*{argmin}_{\boldsymbol{\alpha}_S \in \mathbb{R}^{m \times k}} (q_S^t)^*(\boldsymbol{\alpha}_S) - \langle \boldsymbol{\alpha}_S, \boldsymbol{f}_S^t \rangle + \frac{\gamma}{2m} \|(J_S^t)^* \boldsymbol{\alpha}_S\|_2^2,$$

for $(q_S^t)^*(\boldsymbol{\alpha}_S) - \langle \boldsymbol{\alpha}_S, \boldsymbol{f}_S^t \rangle = \langle \boldsymbol{g}_S^t - \boldsymbol{\alpha}_S, H_S^{-1}(\boldsymbol{g}_S^t - \boldsymbol{\alpha}_S) \rangle$, where we used the inverse-Hessian-vector product

$$H_S^{-1} \boldsymbol{\alpha}_S := (\nabla \ell_{i_1}^2 (\boldsymbol{f}_{i_1}^t)^{-1} \boldsymbol{\alpha}_1, \dots, \nabla \ell_{i_m}^2 (\boldsymbol{f}_{i_m}^t)^{-1} \boldsymbol{\alpha}_m).$$

For the **squared loss**, we naturally get back the solution presented earlier. For the **logistic loss**, while the Hessian is positive semi-definite, it is not invertible, as we have $\nabla^2 \ell_i(\boldsymbol{f}_i^t)^\top \mathbf{1}_k = 0$ for all $i \in [n]$.

Generally for any positive-semi-definite Hessian, the dual can still be formulated as an **equality-constrained QP**, see Appendix C.3. The direction can be computed as $\boldsymbol{d}_S^t \approx \frac{\gamma}{m}(J_S^t)^* \boldsymbol{\alpha}_S^t$, with $\boldsymbol{\alpha}_S^t = \boldsymbol{g}_S^t - \boldsymbol{\beta}_S^t$ and

$$\boldsymbol{\beta}_S^t \approx \operatorname*{argmin}_{\boldsymbol{\beta} \in \mathbb{R}^{m \times k}} \frac{1}{2} \langle \boldsymbol{\beta}, (H_S^t)^\dagger \boldsymbol{\beta} \rangle + \frac{\gamma}{2m} \|(J_S^t)^*(\boldsymbol{g}_S^t - \boldsymbol{\beta})\|_2^2$$

$$\text{s.t. } (\mathrm{I} - H_S^t (H_S^t)^\dagger) \boldsymbol{\beta} = \mathbf{0}, \tag{8}$$

where we used the pseudo-inverse Hessian product

$$(H_S^t)^\dagger \boldsymbol{\alpha} := ((H_{i_1}^t)^\dagger \boldsymbol{\alpha}_1, \dots, (H_{i_m}^t)^\dagger \boldsymbol{\alpha}_m) \tag{9}$$

The above equality-constrained QP can be solved efficiently with a **conjugate gradient** method, using $H_S^t (H_S^t)^\dagger$ as a preconditioner and initializing at a dual variable respecting the constraints, see Appendix C.3. Note that if the subproblem in $\boldsymbol{\beta}$ is initialized at $\mathbf{0}$ as done in the experiments, the output direction is a gradient at iteration 0. Each subsequent iteration therefore improves on the gradient.

**Example.** For the **logistic loss**, the pseudo-inverse enjoys a **closed form**. The direction can then be computed as $\boldsymbol{d}_S^t \approx \frac{\gamma}{m}(J_S^t)^* \boldsymbol{\alpha}_S^t$, with $\boldsymbol{\alpha}_S^t = \boldsymbol{g}_S^t - \boldsymbol{\beta}_S^t$ for

$$\boldsymbol{\beta}_S^t \approx \operatorname*{argmin}_{\boldsymbol{\beta} \in \mathbb{R}^{m \times k}} \frac{1}{2}\langle \boldsymbol{\beta}, D_S^{-1}\boldsymbol{\beta}\rangle + \frac{\gamma}{2m}\|(J_S^t)^*(\boldsymbol{g}_S^t - \boldsymbol{\beta})\|_2^2$$

$$\text{s.t. } \mathbf{1}_k^\top \boldsymbol{\beta}_i = \mathbf{0}, \ i \in [m]. \tag{10}$$

Here, denoting $\sigma$ the softmax, we defined

$$D_S^{-1}\boldsymbol{\beta} := (\boldsymbol{\beta}_1/\sigma(\boldsymbol{f}_{i_1}^t), \ldots, \boldsymbol{\beta}_m/\sigma(\boldsymbol{f}_{i_m}^t)),$$

where division is performed element-wise.

### 3.3 Linear case: connection with SDCA

We now discuss the setting when $f_i$ is linear. In linear multiclass classification, with $k$ classes and $d$ features, we set $f_i(\boldsymbol{w}) = \boldsymbol{W}\boldsymbol{x}_i$, where $\boldsymbol{W} \in \mathbb{R}^{k \times d}$ is a matrix representation of $\boldsymbol{w} \in \mathbb{R}^p$, with $p = kd$. We then have

$$(J_i^t)^* \boldsymbol{\alpha}_i = \partial f_i(\boldsymbol{w}^t)^* \boldsymbol{\alpha}_i = \text{vec}(\boldsymbol{\alpha}_i \boldsymbol{x}_i^\top) \in \mathbb{R}^{kd}.$$

The key difference with the nonlinear $f_i$ setting is that the Jacobian $J_i^t$ is actually independent of $\boldsymbol{w}^t$, so that

$$\|(J_i^t)^* \boldsymbol{\alpha}_i\|_2^2 = \langle \boldsymbol{\alpha}_i, \|\boldsymbol{x}_i\|_2^2 \, \mathrm{I} \, \boldsymbol{\alpha}_i\rangle = \|\boldsymbol{x}_i\|_2^2 \cdot \|\boldsymbol{\alpha}_i\|_2^2.$$

Contrary to the setting where $f_i$ is nonlinear, the Hessian of $\|(J_i^t)^* \boldsymbol{\alpha}_i\|_2^2$ is diagonal, which makes the subproblem easier to solve. Indeed, when the batch size is $m = 1$, denoting $\sigma_i := (\gamma\|\boldsymbol{x}_i\|_2^2)^{-1}$, we arrive at the dual subproblem

$$\boldsymbol{\alpha}(\gamma\ell_i, f_i)(\boldsymbol{w}^t) = \operatorname*{argmin}_{\boldsymbol{\alpha}_i \in \mathbb{R}^k} \ell_i^*(\boldsymbol{\alpha}_i) - \langle \boldsymbol{\alpha}_i, \boldsymbol{f}_i^t\rangle + \frac{1}{2\sigma_i}\|\boldsymbol{\alpha}_i\|_2^2$$

$$= \text{prox}_{\sigma_i \ell_i^*}(\sigma_i \boldsymbol{f}_i^t).$$

This is exactly the dual subproblem used in SDCA (Shalev-Shwartz & Zhang, 2013). It enjoys a closed form in the case of the squared, hinge and sparsemax loss functions (Blondel et al., 2020). When the batch size is $m$, we obtain that the dual subproblem solution $\boldsymbol{\alpha}(\gamma\ell_S, f_S)(\boldsymbol{w}^t)$ is equal to

$$\operatorname*{argmin}_{\boldsymbol{\alpha}_S \in \mathbb{R}^{m \times k}} \ell_S^*(\boldsymbol{\alpha}_S) - \langle \boldsymbol{\alpha}_S, \boldsymbol{f}_S^t\rangle + \frac{\gamma}{2m}\langle \boldsymbol{\alpha}_S, \boldsymbol{K}\boldsymbol{\alpha}_S\rangle,$$

where we defined the kernel matrix $[\boldsymbol{K}]_{i,j} := \langle \boldsymbol{x}_i, \boldsymbol{x}_j\rangle$. There is no closed form in this case.

## 4 Analysis

We now review prox-linear directions theoretically.

**Approximation error.** When $\ell_i$ is $C_\ell$-Lipschitz continuous and $f_i$ is $L_f$-smooth, the partial linearization of $h_i := \ell_i \circ f_i$ satisfies a quadratic approximation error (Drusvyatskiy & Paquette, 2019, Lemma 3.2), for all $\boldsymbol{w}, \boldsymbol{u} \in \mathbb{R}^p$,

$$|h_i(\boldsymbol{w} + \boldsymbol{u}) - \ell_i(f_i(\boldsymbol{w}) + \partial f_i(\boldsymbol{w})\boldsymbol{u})| \le \frac{C_\ell L_f}{2}\|\boldsymbol{u}\|_2^2.$$

In comparison, if in addition, $\ell_i$ is $L_\ell$ smooth and $f_i$ is $C_f$-Lipschitz continuous, a linear approximation of $h_i$ has a quadratic error of the form

$$|h_i(\boldsymbol{w} + \boldsymbol{u}) - h_i(\boldsymbol{w}) - \nabla h_i(\boldsymbol{w})\boldsymbol{u}| \le \frac{C_\ell L_f + C_f^2 L_\ell}{2}\|\boldsymbol{u}\|_2^2.$$

The above result confirms that, unsurprisingly, the partial linearization is theoretically a more accurate approximation than the full linearization.

---

**Algorithm 2** Dual-based prox-linear direction

---

1: **Inputs:** network outputs $\boldsymbol{f}_S^t$, JVP $J_S^t$ and VJP $(J_S^t)^*$ as in Algorithm 1, regularization $\gamma$ (1 if linesearch is used)

2: Run inner solver to approximately solve (7)

$$\boldsymbol{\alpha}_S^t \approx \underset{\boldsymbol{\alpha}_S \in \mathbb{R}^{m \times k}}{\operatorname{argmin}} \; \ell_S^*(\boldsymbol{\alpha}_S) - \langle \boldsymbol{\alpha}_S, \boldsymbol{f}_S^t \rangle + \frac{\gamma}{2m} \|(J_S^t)^* \boldsymbol{\alpha}_S\|_2^2,$$

or its quadratic approximation

$$\boldsymbol{\alpha}_S^t \approx \underset{\boldsymbol{\alpha}_S \in \mathbb{R}^{m \times k}}{\operatorname{argmin}} \; (q_S^t)^*(\boldsymbol{\alpha}_S) - \langle \boldsymbol{\alpha}_S, \boldsymbol{f}_S^t \rangle + \frac{\gamma}{2m} \|(J_S^t)^* \boldsymbol{\alpha}_S\|_2^2,$$

detailed in (8), and in (10) for the logistic loss.

3: Map back to primal direction

$$\boldsymbol{d}_S^t := \frac{\gamma}{m} (J_S^t)^* \boldsymbol{\alpha}_S^t = \frac{\gamma}{m} \sum_{i \in S} (J_i^t)^* \boldsymbol{\alpha}_i^t$$

4: Set next parameters $\boldsymbol{w}^{t+1}$ by

$$\boldsymbol{w}^{t+1} := \boldsymbol{w}^t - \boldsymbol{d}_S^t \qquad \text{(fixed stepsize (5))}$$

$$\text{or} \quad \boldsymbol{w}^{t+1} := \boldsymbol{w}^t - \eta^t \boldsymbol{d}_S^t \qquad \text{(linesearch (6))}$$

for $\eta^t$ s.t. $h_S(\boldsymbol{w}^t - \eta^t \boldsymbol{d}_S^t) \leq h_S(\boldsymbol{w}^t) - \beta \eta^t \langle \boldsymbol{d}_S^t, \boldsymbol{g}_S^t \rangle$.

5: **Output:** $\boldsymbol{w}^{t+1} \in \mathbb{R}^p$

---

**Descent direction.** To integrate a prox-linear direction $\boldsymbol{d} = \boldsymbol{d}(\gamma \ell_S, f_S)$ or $\boldsymbol{d}(\gamma q_S^t, f_S)$, into generic optimization algorithms, it is preferable if $-\boldsymbol{d}$ defines a **descent direction** w.r.t. the mini-batch stochastic gradient $\nabla h_S(\boldsymbol{w})$, namely, $-\boldsymbol{d}$ should satisfy

$$\langle -\boldsymbol{d}, \nabla h_S(\boldsymbol{w}) \rangle \leq 0. \tag{11}$$

Thanks to the convexity of the subproblem, we can show that the exact prox-linear direction satisfies (11).

**Proposition 1.** *If each $\ell_i$ is convex, the negative direction $-\boldsymbol{d}(\gamma \ell_S, f_S)(\boldsymbol{w}^t)$ or its quadratic approximation direction $-\boldsymbol{d}(\gamma q_S^t, f_S)(\boldsymbol{w}^t)$ define descent directions for the composition $h_S = \ell_S \circ f_S$ at $\boldsymbol{w}^t$.*

In practice, we never compute the exact direction. We show below that the approximate directions obtained by the conjugate gradient method run in the primal or in the dual define descent directions for any number of iterations. Proofs are presented in Appendix C.6.

**Proposition 2.** *Let $\boldsymbol{d}_S^{t;\tau}$ be the direction obtained after $\tau$ iterations of the conjugate gradient method, for solving line 4 of Algo. 1 (primal) or line 2 of Algo. 2, followed by line 3 (dual). Then $-\boldsymbol{d}_S^{t;\tau}$ is a descent direction for $h_S = \ell_S \circ f_S$ at $\boldsymbol{w}^t$.*

**Computational cost.** The computational costs associated to the resolution of the inner subproblems in the primal and dual formulations depend on (i) the cost of each iteration of the inner solver, (ii) the convergence rate associated to the subproblem. For quadratic inner subproblems, arising for example with a squared loss, we detail in Appendix D that the cost of running $\tau$ inner iterations of CG in the primal or the dual are respectively

$$\text{Primal: } \tau(\mathcal{T}_{\text{JVP}-\text{VJP}} + O(p)),$$

$$\text{Dual: } \tau(\mathcal{T}_{\text{JVP}-\text{VJP}} + O(mk)),$$

where $\mathcal{T}_{\text{JVP}-\text{VJP}}$ denotes the computational complexity of a call to a JVP $\partial \boldsymbol{f}_S(\boldsymbol{w})$ and a call to a VJP $\partial \boldsymbol{f}_S(\boldsymbol{w})^*$. Therefore, the dual formulation, which operates in the dual space leads to less expansive inner updates. In addition, the convergence rate of CG on the inner subproblems depend theoretically on their

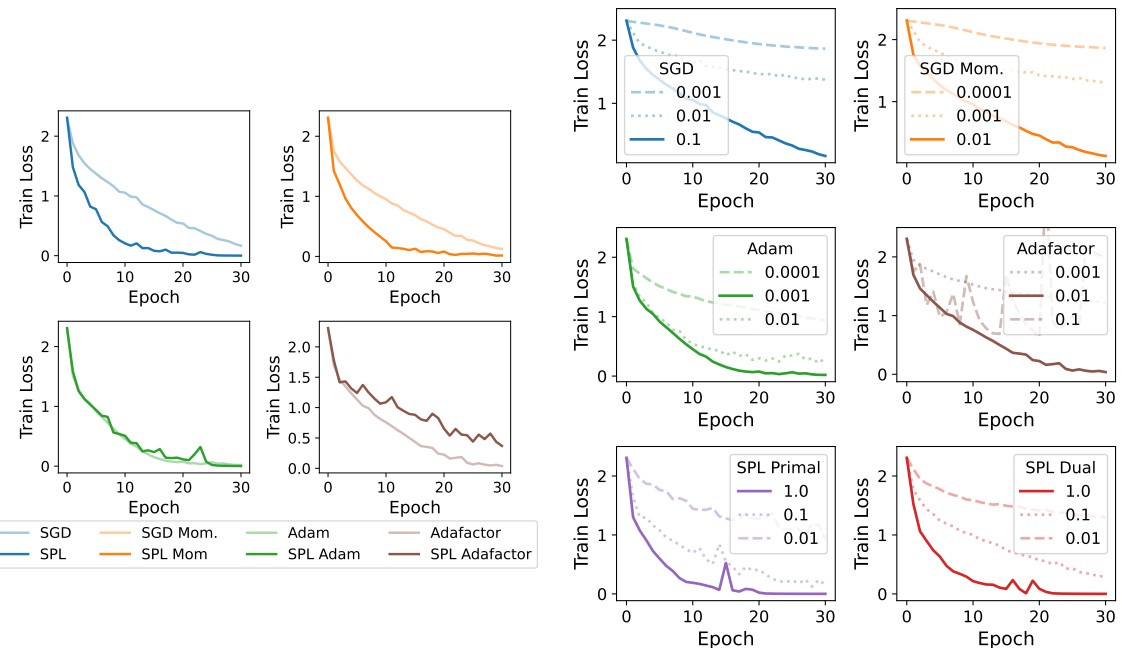

Figure 1: Benefit of using prox-linear directions as replacement for stochastic gradients in existing solvers (CIFAR10, ConvNet).

Figure 2: Robustness to stepsize (CIFAR10, ConvNet).

condition numbers and the distribution of eigenvalues of the operators considered as detailed in Appendix D. Since we consider in practice only few inner iterations, the convergence rate of the method has much less influence than the per-iteration cost.

## 5  Experiments

We consider image classification tasks with a **logistic loss** using a prox-linear direction approximated via the dual formulation of the quadratic approximation of the loss, using a conjugate gradient method with 2 inner iterations. We use iterates of the form $\boldsymbol{w}^{t+1} = \boldsymbol{w}^t - \boldsymbol{d}(\gamma^t q_S^t, f_S)(\boldsymbol{w}^t)$ denoted **SPL** for **stochastic prox-linear** and iterates of the form $\boldsymbol{w}^{t+1} = \boldsymbol{w}^t - \eta^t \boldsymbol{d}(q_S^t, f_S)(\boldsymbol{w}^t)$ for $\eta^t$ chosen by an Armijo line-search, denoted **Armijo SPL**.

For preliminary diagnosis, we compare the performance of Armijo SPL to several stochastic gradients based optimization schemes to classify images from the CIFAR10 dataset (Krizhevsky & Hinton, 2009) with a three layer ConvNet presented in Appendix E in terms of epochs. Stepsizes are searched on a log10 scale with a batch-size 256.

### 5.1  Prox-linear vs. stochastic gradient

In Figure 1, we employ the prox-linear direction as a replacement for the stochastic gradient in existing algorithms, ranging from ADAM to SGD with momentum or AdaFactor. Namely, we simply replace $\nabla h_S(\boldsymbol{w}^t)$ by $\boldsymbol{d}(q_S^t, f_S)(\boldsymbol{w}^t)$ in each solver's update rule. Results in time are in Appendix E.

We observe that for most update rules, SPL generally speeds up the convergence or performs on par in terms of epochs, except for AdaFactor for which using prox-linear directions perform similarly to gradients.

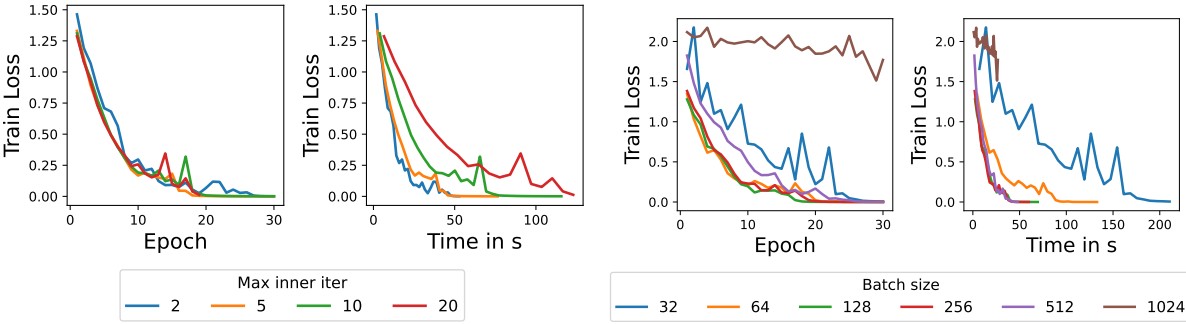

Figure 3: Robustness to #inner iterations (top) and batch size (bottom), on CIFAR10 with a ConvNet. Left: epochs. Right: wallclock time.

## 5.2 Dual vs. primal

In Figure 4, we compare the runtime performance of the prox-linear direction, depending on whether the primal or the dual was used. In both cases, we use an Armijo-SPL. While both approaches perform similarly in terms of iterations, we observe that the dual approach brings some gains in total runtime.

## 5.3 Robustness to hyperparameters

We analyze the sensitivity to hyperparameters of algorithms based on the prox-linear direction. Our goal here is to understand the trade-offs involved in using prox-linear directions.

**Robustness to stepsize.** We study in Figure 2 the robustness to the stepsize selection of different algorithms. For each algorithm, we display in solid line the best stepsize found among $(10^i)_{i=-4}^0$ and in transparent lines the two other best stepsizes. Here we use prox-linear udpates with varying "inner-stepsize" $\gamma$ to analyze the benefits of incorporating the stepsize inside the oracle with $\gamma$ chosen in $(10^i)_{i=-4}^0$. Namely, here we consider updates $\boldsymbol{w}^{t+1} = \boldsymbol{w}^t - \boldsymbol{d}(\gamma q_S^t, f_S)(\boldsymbol{w}^t)$, denoted **SPL**, rather than $\boldsymbol{w}^{t+1} = \boldsymbol{w}^t - \eta^t \boldsymbol{d}(q_S^t, f_S)(\boldsymbol{w}^t)$.

We observe that SPL provides competitive performance for a larger range of stepsizes than other algorithms such as SGD, SGD with momentum, or AdaFactor, while exhibiting a similar robustness as Adam.

**Robustness to number of inner iterations and batch size.** One of the hyperparameters of the algorithm is the number of inner iterations. On the left panel of Figure 3, we analyzed the behavior of **Armijo SPL** when varying the number of inner iterations. We did not observe an important sensitivity in this setting.

On the right panel Figure 3, we observe that for too small or too large mini-batch sizes, **Armijo SPL** does not perform as well as for medium sizes. Indeed, if the batch is too small, the variance may be too big. On the other hand, if the batch is too big, since the batch-size influences the conditioning of subproblem, making only a few steps of the subroutine may not be sufficient to get a good direction.

## 5.4 Comparison with existing algorithms

Initial comparisons in Figure 1 demonstrated the benefits of SPL in terms of epochs. In Figure 5, we compare the performance of Armijo SPL to several stochastic gradient based optimization on the same ConvNet on CIFAR10 in terms of wallclock time. In this experiment, Armijo SPL and SPL are able to reach higher test accuracy overall. However, even with only two inner CG steps, SPL remains twice longer to reach convergence.

Additional experiments on ResNets on Imagenet and various ConvNets on CIFAR10 are presented in Appendix F. These additional experiments confirm with the ones presented here: while our results are promising, adaptive optimizers like ADAM remain very competitive thanks to their low computational cost.

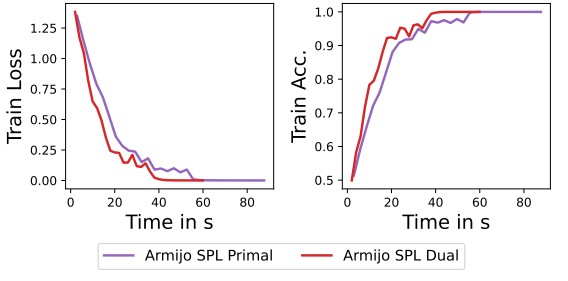
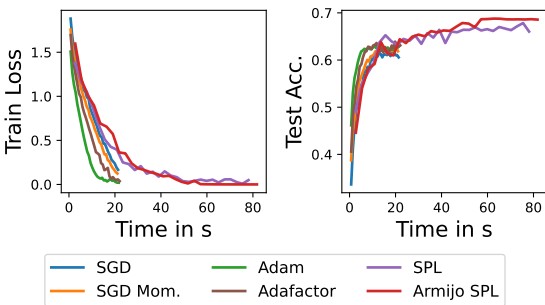

Figure 4: Runtime comparison between primal-based and dual-based computations (CIFAR10, ConvNet).

Figure 5: CIFAR-10 with a ConvNet.

In Appendix F, we show that our approach is competitive with Shampoo and KFAC. In particular, for small batch sizes, our approach is faster than KFAC in time. The sensitivity of such methods to the batch size remains a challenge, to make them more widely applicable.

## 6 Conclusion

Prox-linear (a.k.a. modified Gauss-Newton) directions can be thought as a **principled middle ground** between the stochastic gradient and the stochastic (regularized) Newton direction. In this paper, we derived the **Fenchel-Rockafellar dual** associated with the corresponding subproblem, which to our knowledge had not been studied before. To solve the dual when a quadratic approximation of the loss is used, we proposed a **conjugate gradient** algorithm, that integrates seamlessly with autodiff through the use of **linear operators** and can handle **equality constraints**. We proved that this algorithm produces **descent directions**, when run for **any number** of inner iterations. Empirically, we found that prox-linear directions work best with **medium batch sizes** and are more **robust** than stochastic gradients to **stepsize specification**.

While we demonstrated promising results, first-order adaptive methods such as ADAM remain an excellent trade-off between accuracy and computational cost. Further work is needed to reduce the computational cost of Gauss-Newton methods in general and of our dual approach in particular. We hope this study brings new insights and is a first step in that direction. To conclude, we point out that our approach can be easily extended to incorporate a **non-differentiable regularizer**, enabling the use of sparsity-inducing penalties on the network weights, as studied in Appendix B.

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

# Dual Gauss-Newton Directions for Deep Learning

## Appendix

Appendix A expands on related work. Appendix B presents how to incorporate additional regularizers into the computation of the Gauss-Newton direction. Appendix C contains all proofs. Appendix D details the computational costs. Finally, Appendix E presents some additional experiments and experimental details.

## A    Detailed literature review

**Variational perspective on optimization oracles.**    The variational perspective on optimization oracles is well-known, see, e.g., Parikh et al. (2014); Beck (2017). This perspective can be used together with full linearization to motivate SGD variants based on Polyak stepsize (Gower et al., 2022).

In addition to the interpretation of prox-linear/Gauss-Newton as an intermediate oracle between gradient and Newton updates, the prox-linear can be interpreted through the lens of proximal point methods. With the notations of Section 1, the proximal point algorithm (Rockafellar, 1976) computes the next iterate as

$$
\boldsymbol{w}^{t+1} \coloneqq \mathrm{prox}(\gamma h_i)(\boldsymbol{w}^t)
$$

$$
\coloneqq \operatorname*{argmin}_{\boldsymbol{w} \in \mathbb{R}^p} h_i(\boldsymbol{w}) + \frac{1}{2\gamma} \|\boldsymbol{w} - \boldsymbol{w}^t\|_2^2
$$

$$
= \boldsymbol{w}^t - \operatorname*{argmin}_{\boldsymbol{d} \in \mathbb{R}^p} h_i(\boldsymbol{w}^t - \boldsymbol{d}) + \frac{1}{2\gamma} \|\boldsymbol{d}\|_2^2,
$$

where we used the change of variable $\boldsymbol{w} \coloneqq \boldsymbol{w}^t - \boldsymbol{d}$. Unfortunately, when $h_i = \ell_i \circ f_i$ and $f_i$ is a neural network, this subproblem is nonconvex. In comparison, we can write the prox-linear update as

$$
\mathrm{prox\_linear}(\gamma \ell_i, f_i)(\boldsymbol{w}^t) \coloneqq \mathrm{prox}(\mathrm{plin}(\gamma \ell_i, f_i, \boldsymbol{w}^t))(\boldsymbol{w}^t)
$$

$$
= \operatorname*{argmin}_{\boldsymbol{w} \in \mathbb{R}^p} \mathrm{plin}(\ell_i, f_i, \boldsymbol{w}^t)(\boldsymbol{w}) + \frac{1}{2\gamma} \|\boldsymbol{w}^t - \boldsymbol{w}\|_2^2
$$

$$
= \boldsymbol{w}^t - \boldsymbol{d}(\gamma \ell_i, f_i)(\boldsymbol{w}^t) \approx \mathrm{prox}(\gamma h_i)(\boldsymbol{w}^t).
$$

Therefore, we can see the resulting update as the proximal point iteration on a partially linearized function, hence the name prox-linear. Importantly however, unlike the proximal point update, the associated subproblem is convex.

**Nonlinear least-squares (deterministic case).**    The idea of exploiting the compositional structure of an objective to partially linearize the objective and minimize the resulting simplified subproblem, stems from the resolution of nonlinear least-squares problems (Björck, 1996) of the form

$$
\min_{\boldsymbol{w} \in \mathbb{R}^p} \|f(\boldsymbol{w})\|_2^2,
$$

where $f$ is a nonlinear differentiable function. In this context, the Gauss-Newton algorithm (Kelley, 1999) proceeds originally by computing

$$
\boldsymbol{d}^t \coloneqq \operatorname*{argmin}_{\boldsymbol{v} \in \mathbb{R}^p} \|f(\boldsymbol{w}^t) - \partial f(\boldsymbol{w}^t)\boldsymbol{d}\|_2^2,
$$

and computing $\boldsymbol{w}^{t+1}$ by means of a line-search along the direction $-\boldsymbol{d}^t$ (which is guaranteed to be a descent direction). The Gauss-Newton algorithm can converge locally to a solution at a quadratic rate provided that the initial point is close enough to the solution (Kelley, 1999). The Gauss-Newton algorithm can be subject to numerical instability as soon as the operator $\partial f(\boldsymbol{w}^t)$ is non-singular. To circumvent this issue, Levenberg (1944) and Marquardt (1963) introduced the now called Levenberg-Marquardt algorithm that computes directions according to a regularized version of the Gauss-Newton direction

$$
\boldsymbol{d}^t \coloneqq \operatorname*{argmin}_{\boldsymbol{d} \in \mathbb{R}^p} \|f(\boldsymbol{w}^t) - \partial f(\boldsymbol{w}^t)\boldsymbol{d}\|_2^2 + \frac{\lambda}{2} \|\boldsymbol{d}\|_2^2.
$$

A line-search along $-\boldsymbol{d}^t$ is then taken to define the next iterate (the direction $-\boldsymbol{d}^t$ is again a descent direction). The parameter $\lambda$ (equivalent to $1/\gamma$ in the derivation of our algorithm) acts as a regularization to ensure that the direction is relevant at the current iterate: larger $\lambda$ induce smaller directions, closer to a negative gradient direction. The parameter $\lambda$ may be modified along the iterations by a trust-region mechanism (see e.g. Bergou et al. (2020)). Namely, a trust-region mechanism computes

$$r_t := \frac{\|f(\boldsymbol{w}^t)\|_2^2 - \|f(\boldsymbol{w}^t - \boldsymbol{d}^t)\|_2^2}{m_t(\boldsymbol{0}) - m_t(\boldsymbol{d}^t)},$$

where $m_t(\boldsymbol{d}) := \|f(\boldsymbol{w}^t) - \partial f(\boldsymbol{w}^t)\boldsymbol{d}\|_2^2 + \lambda_t\|\boldsymbol{d}\|_2^2/2$ the approximate model of the objective. Here, $r_t$ is a measure of how good the model was for the current $\lambda_t$. For $r_t \gg 0$, the step taken reduced efficiently the original objective, while for, e.g., $r_t \ll 0$, the model provided a bad direction. At each iteration, if $r_t > \delta_1$, $\lambda_{t+1} = 0.5\lambda_t$ for example, is decreased, if $\rho_t < \delta_2$, the iteration is a priori redone with an increased $\lambda_t$, or we may simply set $\lambda_{t+1} = 2\lambda_t$ for the next iteration. Gauss-Newton-like algorithms have been applied succesfully to phase-retrieval (Herring et al., 2019; Repetti et al., 2014), nonlinear control (Sideris & Bobrow, 2005), non-negative matrix factorization (Huang & Fu, 2019) to cite a few.

**Compositional problems (deterministic case).** Gauss-Newton-like algorithms have then been generalized beyond nonlinear least-squares to tackle generic compositional problems of the form $\ell \circ f + \rho$ with $\ell$ convex, $f$ differentiable nonlinear, $\rho$ a simple function with computable proximity operator. The resulting algorithm, called prox-linear (Burke, 1985) computes the next oracle as

$$\boldsymbol{w}^{t+1} := \operatorname*{argmin}_{\boldsymbol{w} \in \mathbb{R}^p} \ell(f(\boldsymbol{w}^t) + \partial f(\boldsymbol{w}^t)(\boldsymbol{w} - \boldsymbol{w}^t)) + \rho(\boldsymbol{w}) + \frac{\lambda}{2}\|\boldsymbol{w} - \boldsymbol{w}^t\|_2^2.$$

Nesterov (2007) proposed to minimize nonlinear residuals with a generic sharp metric such as $\ell = \|\cdot\|_2$ with $\rho = 0$. Nesterov (2007) proved global convergence rates of the above method given that $\sigma_{\min}(\partial f(\boldsymbol{w})^*) \geq \sigma > 0$ for any $\boldsymbol{w}$ and also gave local convergence rates for $\sigma_{\min}(\partial f(\boldsymbol{w})) \geq \sigma > 0$ around an initial point close to a local solution. Drusvyatskiy & Paquette (2019) considered prox-linear algorithms for finite sum objectives, that is, problems of the form $\frac{1}{n}\sum_{i=1}^n \ell_i \circ f_i + \rho$. Drusvyatskiy & Paquette (2019) considered the norm of the scaled difference between iterates as a stationary measure of the algorithm by relating it to the gradient of the Moreau envelope of the objective. They proposed to solve the sub-problem up to a near-stationarity criterion defined by the norm of the (sub)-gradient of the dual objective of the sub-problem. They derived the total computational complexity of the algorithm when using various inner solvers from an accelerated gradient algorithm to fast incremental solvers such as SVRG or its accelerated version. Rates are provided for the case where $\ell$ is smooth or non-smooth but smoothable. Pillutla et al. (2019) evaluated these algorithms in the context of structured prediction with smoothed oracles. The reported performance was on par with SGD. Pillutla et al. (2023) performed more synthetic experiments with similar conclusions, casting doubt on the usefulness of the method. Note however that Pillutla et al. (2019; 2023) did not consider varying the regularization, nor a mini-batch version of the algorithm, nor incorporating the algorithm into other first-order mechanisms.

For compositional problems with general loss $\ell$, an alternative to the prox-linear algorithm is to use a quadratic approximation of $\ell$ together with the linearization of $f$ (Messerer et al., 2021; Roulet et al., 2022). Roulet et al. (2022) showed the global convergence and local convergence of such a method with the same assumption as Nesterov (2007), i.e., $\sigma_{\min}(\partial f(\boldsymbol{w})^*) \geq \sigma > 0$, while asserting this condition in some nonlinear control problems. Diehl & Messerer (2019) considered the local convergence of generalized Gauss-Newton algorithms under suitable assumptions.

**Stochastic case.** Duchi & Ruan (2018) studied the asymptotic convergence of stochastic versions of the prox-linear algorithms. They considered objectives of the form $\mathbb{E}_{z \sim p}[\ell(f(\cdot; z); z)] + \rho$ for some unknown distribution $p$. The setting considered in our paper is an instance of such a problem. The prox-linear algorithm consists then in iterates of the form

$$\boldsymbol{w}^{t+1} = \operatorname*{argmin}_{\boldsymbol{w} \in \mathbb{R}^p} \frac{1}{|S|}\sum_{i \in S} \ell(f(\boldsymbol{w}^t; z_i) + \partial f(\boldsymbol{w}^t; z_i)(\boldsymbol{w} - \boldsymbol{w}^t); z_i) + \rho(\boldsymbol{w}) + \frac{\lambda}{2}\|\boldsymbol{w} - \boldsymbol{w}^t\|_2^2,$$

for $(z_i)_{i \in S} \overset{i.i.d.}{\sim} p$ a mini-batch of samples. Duchi & Ruan (2018) presented asymptotic rates of convergence for this method with experiments on phase retrieval problems. In a slightly different spirit, namely, objectives of the form $\ell(\mathbb{E}_{z \sim p}[f(\cdot; z)]) + \rho$, Tran-Dinh et al. (2020); Zhang & Xiao (2021) presented convergence rates using estimators of the Jacobian and the function values of $f$, with stochastic estimators such as SPIDER, SARAH or simply large mini-batches. Gargiani et al. (2020) considered stochastic versions of the generalized Gauss-Newton algorithm for deep learning applications with simple experiments on MNIST, FashionMNIST an CIFAR10. However, they did not use the dual as we do, and they did not consider using the oracle as a replacement for the gradient in existing solvers such as ADAM.

**Jacobian/Hessian-free Gauss-Newton methods in deep learning.**   To harness the potential power of second-order optimization algorithms such as a Newton method, Martens (2010) considered implementing a Newton method by accessing Hessian-vector products and inverting the Hessian by a conjugate gradient method. Martens (2010) introduced usual techniques from the Newton method such as damping and trust-region techniques to tune the regularization. An issue quickly pointed out by Martens (2010) is that the Hessian of the network need not be positive definite which means that $H^{-1}g$ for $H$ the Hessian of the objective and $g$ the gradient does not necessarily lead to a descent direction. While the Hessian could be regularized to prevent this issue, this may add a non-negligible overhead. In contrast, oracles based on partial linearizations à la Gauss-Newton provably return a descent direction. Therefore, Martens (2010) considered in practice a Gauss-Newton algorithm. Martens (2010) also considered some preconditioning techniques for the conjugate gradient method and argues for large mini-batch sizes, see Section 4 of the aforementioned paper. Martens & Sutskever (2011) applied this technique to recurrent neural networks.

Recently, Ren & Goldfarb (2019) revised such a technique by solving the subproblem associated to a generalized Gauss-Newton iteration by means of the Woodbury formula and using a form of trust-region technique. The Woodbury formula aims to reduce computational costs associated to inversions by exploiting the structure of the subproblems. In the case of the squared loss, the two approaches coincide. However, for the quadratic approximation of the logistic loss, Ren & Goldfarb (2019) considered a non-symmetric development of the Woodbury formula, that differs from our approach. By carefully tackling the dual formulation of the problem in the case of a quadratic approximation of the logistic loss, we are able to keep the same inner solver, i.e., a conjugate gradient method, and to prove that the resulting direction is always a descent direction. Ren & Goldfarb (2019) performed experiments on CIFAR10, MNIST and webspam with two hidden layers MLPS. In a similar spirit, Henriques et al. (2019) considered a Hessian-free optimization algorithm to compute exactly a Newton step. Henriques et al. (2019) considered performing one gradient step on the subproblem by means of a Hessian-vector product and use the resulting direction in place of the gradient. The algorithm of Henriques et al. (2019) can therefore be seen as an extreme case of our algorithm with only one iteration in the subproblem (with the additional difference that they consider a Newton step). Henriques et al. (2019) conducted an extensive set of experiments on CIFAR10 with ConvNets, ResNets, ImageNet with VGG ConvNet and MNIST with MLPS as well as standard difficult nonconvex objectives such as the Rosenberg function. They observed some gains in accuracy and speed in epochs. On the other hand, gains in time were only reported for a small architecture (Henriques et al., 2019, Figure 3).

**Approximating the Gauss-Newton matrix by block diagonal blocks.**   Rather than solving approximately for a Gauss-Newton-like direction, a line of work starting from Martens & Grosse (2015) considered using a block diagonal approximation of the Gauss-Newton matrix. Such a direction often took the terminology of natural gradient descent algorithms since, for losses stemming from an exponential family, a generalized Gauss-Newton method coincides with a natural gradient descent (Martens, 2020). The resulting algorithm called KFAC (Kronecker Factored Approximate Curvature) stems from the observation that the block of the Gauss-Newton matrix corresponding to the $k^{\text{th}}$ can be factorized as a Kronecker product whose expectation may be approximated as the Kronecker product of the expectations. George et al. (2018) extended the KFAC method in an algorithm EKFAC that further tries to compute an adequate eigenbasis along which to compute the approximate blocks. Botev et al. (2017) considered a finer decomposition of the block diagonal Gauss-Newton matrix computed by back-propagating the information through the graph of a feed-forward network leading to the algorithm KFRA. Both KFAC and KFRA a priori depend on the proposed architecture. They were developed for MLPs and subsequently extended to convolutional neural networks (Pauloski et al.,

2020) and transformers architecture (Zhang et al., 2022). Such layer-wise decomposition of second-order methods is reminiscent of techniques used in nonlinear control to implement Gauss-Newton methods or their nonlinear control variant called differentiable dynamic programming which were adapted to a deep learning context by Liu et al. (2020). Recently, Gupta et al. (2018); Anil et al. (2020) generalized the idea of computing generic preconditioners for deep networks by exploiting their tensor structure.

# B   Non-differentiable regularizer extension

In this section, we consider regularized objectives of the form

$$\min_{\boldsymbol{w}\in\mathbb{R}^p}\left[\frac{1}{n}\sum_{i=1}^n h_i(\boldsymbol{w}) + \rho(\boldsymbol{w}) = \frac{1}{n}\sum_{i=1}^n \ell_i(f_i(\boldsymbol{w})) + \rho(\boldsymbol{w})\right],$$

where $\rho$ is a potentially non-differentiable regularization, such as the sparsity-inducing penalty $\rho(\boldsymbol{w}) = \lambda\|\boldsymbol{w}\|_1$, where $\lambda > 0$ controls the regularization strength.

## B.1   Primal

In order to perform an update on a mini-batch $S$, we can solve

$$\boldsymbol{w}(\gamma\ell_S, f_S, \gamma\rho)(\boldsymbol{w}^t) := \operatorname*{argmin}_{\boldsymbol{w}\in\mathbb{R}^p} \frac{1}{m}\sum_{i\in S}\mathrm{plin}(\ell_i, \boldsymbol{f}_i, \boldsymbol{w}^t)(\boldsymbol{w}) + \frac{1}{2\gamma}\|\boldsymbol{w}^t - \boldsymbol{w}\|_2^2 + \rho(\boldsymbol{w})$$

$$= \operatorname*{argmin}_{\boldsymbol{w}\in\mathbb{R}^p} \sum_{i\in S}\ell_i(\boldsymbol{f}_i^t + J_i^t(\boldsymbol{w} - \boldsymbol{w}^t)) + m\left[\frac{1}{2\gamma}\|\boldsymbol{w}^t - \boldsymbol{w}\|_2^2 + \rho(\boldsymbol{w})\right].$$

Using the change of variable $\boldsymbol{w} := \boldsymbol{w}^t - \boldsymbol{d}$, this can equivalently be written

$$\boldsymbol{w}(\gamma\ell_S, f_S, \gamma\rho)(\boldsymbol{w}^t) = \boldsymbol{w}^t - \boldsymbol{d}(\gamma\ell_S, f_S, \gamma\rho)(\boldsymbol{w}^t),$$

where

$$\boldsymbol{d}(\gamma\ell_S, f_S, \gamma\rho)(\boldsymbol{w}^t) := \operatorname*{argmin}_{\boldsymbol{d}\in\mathbb{R}^p} \sum_{i\in S}\ell_i(\boldsymbol{f}_i^t - J_i^t\boldsymbol{d}) + m\left[\frac{1}{2\gamma}\|\boldsymbol{d}\|_2^2 + \rho(\boldsymbol{w}^t - \boldsymbol{d})\right] \tag{12}$$

For sparsity-inducing penalties such as the $\ell_1$ norm, this is a non-smooth convex problem that can be solved by a **proximal gradient method**, by using the proximal operator associated to the regularization $\rho$. We rather consider an approximation of the above problem allowing for an algorithm that does not require any additional hyperparameters.

## B.2   Dual

Let us denote $r_t(\boldsymbol{d}) := m\left[\frac{1}{2\gamma}\|\boldsymbol{d}\|_2^2 + \rho(\boldsymbol{w}^t - \boldsymbol{d})\right]$. From Proposition 3, the dual of the primal subproblem (12) is

$$\boldsymbol{\alpha}(\gamma\ell_S, f_S, \gamma\rho)(\boldsymbol{w}^t) := \operatorname*{argmin}_{\boldsymbol{\alpha}_S\in\mathbb{R}^{m\times k}} \ell_S^*(\boldsymbol{\alpha}_S) - \langle \boldsymbol{f}_S^t, \boldsymbol{\alpha}_S\rangle + r_t^*\left((J_S^t)^*\boldsymbol{\alpha}_S\right).$$

Unfortunately, this subproblem may be difficult to solve in general. As shown in Proposition 4, assuming $r_t$ is $\mu$-strongly convex, we can approximate the dual subproblem around any $\boldsymbol{u}^t$ with

$$\boldsymbol{\alpha}(\gamma\ell_S, f_S, \gamma\rho)(\boldsymbol{w}^t) \approx \operatorname*{argmin}_{\boldsymbol{\alpha}_S\in\mathbb{R}^{m\times k}} \ell_S^*(\boldsymbol{\alpha}_S) - \langle \boldsymbol{f}_S^t - \boldsymbol{\delta}_S^t, \boldsymbol{\alpha}_S\rangle + \frac{1}{2\mu}\|(J_S^t)^*\boldsymbol{\alpha}_S\|^2,$$

where $\boldsymbol{\delta}_S^t := J_S^t\left(\nabla r_t^*(\boldsymbol{u}^t) - \frac{1}{\mu}\boldsymbol{u}^t\right)$. We choose $\boldsymbol{u}^t = \boldsymbol{0}$, so that approximating the computation of $\boldsymbol{d}(\gamma\ell_S, f_S, \gamma\rho)(\boldsymbol{w}^t)$ around $\boldsymbol{d} = \boldsymbol{0}$ amounts to approximating the computation of $\boldsymbol{w}(\gamma\ell_S, f_S, \gamma\rho)(\boldsymbol{w}^t)$ around $\boldsymbol{w}^t$. Note that if $\rho(\boldsymbol{w}) = 0$, we get $\boldsymbol{w}^t = \frac{1}{\mu}\boldsymbol{u}^t$, so that $\boldsymbol{\delta}_S^t = \boldsymbol{0}$. Therefore, we recover the dual subproblem (7) in this case.

The entire procedure is summarized in Algorithm 3.

---

**Algorithm 3** Dual-based prox-linear algorithm with $\mu$-strongly convex regularization $r_t$

1: **Inputs:** Parameters $\boldsymbol{w}^t$, "inner stepsize" $\gamma > 0$
2: Compute network outputs $\boldsymbol{f}_S^t$, instantiate JVP $J_S^t$ and VJP $(J_S^t)^*$ as in Algorithm 1
3: Compute for any $\boldsymbol{u}^t$, e.g., $\boldsymbol{u}^t := \boldsymbol{0}$,

$$\boldsymbol{\delta}_S^t := J_S^t \left( \nabla r_t^*(\boldsymbol{u}^t) - \frac{1}{\mu}\boldsymbol{u}^t \right)$$

4: Run inner solver to approximately solve

$$\boldsymbol{\alpha}_S^t \approx \underset{\boldsymbol{\alpha}_S \in \mathbb{R}^{m \times k}}{\operatorname{argmin}} \; \ell_S^*(\boldsymbol{\alpha}_S) - \langle \boldsymbol{\alpha}_S, \boldsymbol{f}_S^t - \boldsymbol{\delta}_S^t \rangle + \frac{1}{2\mu}\|(J_S^t)^*\boldsymbol{\alpha}_S\|_2^2$$

or to approximately solve the equality constrained QP

$$\boldsymbol{\alpha}_S^t \approx \underset{\boldsymbol{\alpha} \in \mathbb{R}^{m \times k}}{\operatorname{argmin}} \frac{1}{2} \langle (\boldsymbol{\alpha}_S - \boldsymbol{g}_S^t), (H_S^t)^\dagger (\boldsymbol{\alpha}_S - \boldsymbol{g}_S^t) \rangle + \langle \boldsymbol{\alpha}_S, \boldsymbol{\delta}_S^t \rangle + \frac{1}{2\mu}\|(J_S^t)^*\boldsymbol{\alpha}_S\|_2^2$$
$$\text{s.t. } (\mathrm{I} - H_S^t(H_S^t)^\dagger)(\boldsymbol{\alpha}_S - \boldsymbol{g}_S^t) = \boldsymbol{0},$$

with $\boldsymbol{g}_S^t = \nabla \ell_S(\boldsymbol{f}_S^t)$, $H_S^t := \nabla^2 \ell_S(\boldsymbol{f}_S^t)$ and $(H_S^t)^\dagger$ the pseudo-inverse (9) (closed forms available for the logistic loss).
5: Compute direction

$$\boldsymbol{d}_S^t := \nabla r_t^*((J_S^t)^*\boldsymbol{\alpha}_S^t)$$

6: Set next parameters $\boldsymbol{w}^{t+1}$ by

$$\boldsymbol{w}^{t+1} := \boldsymbol{w}^t - \boldsymbol{d}_S^t \quad \text{(fixed stepsize (5))} \qquad \text{or} \qquad \boldsymbol{w}^{t+1} := \boldsymbol{w}^t - \eta^t \boldsymbol{d}_S^t \quad \text{(linesearch (6))}$$

for $\eta^t$ s.t. $h_S(\boldsymbol{w}^t - \eta^t \boldsymbol{d}_S^t) \le h_S(\boldsymbol{w}^t) - \beta \eta^t \langle \boldsymbol{d}_S^t, \boldsymbol{g}_S^t \rangle$.
7: **Outputs:** $\boldsymbol{w}^{t+1}$

---

### B.3 Examples of regularizers

**Quadratic regularization.** If $r_t(\boldsymbol{d}) = \frac{m}{2\gamma}\|\boldsymbol{d}\|_2^2$, which is strongly convex with constant $\mu = \frac{m}{\gamma}$, we obtain $r_t^*(\boldsymbol{u}) = \frac{\gamma}{2m}\|\boldsymbol{u}\|_2^2$ and therefore $\nabla r_t^*(\boldsymbol{u}) = \frac{\gamma}{m}\boldsymbol{u}$. We therefore recover Algorithm 2.

**Sum of quadratic and another regularization.** If $r_t(\boldsymbol{d}) = \frac{m}{2\gamma}\|\boldsymbol{d}\|_2^2 + m\,\rho(\boldsymbol{w}^t - \boldsymbol{d})$, which is strongly convex with constant $\mu = \frac{m}{\gamma}$ in general and with $\mu = \frac{m}{\gamma} + m\lambda$ if $\rho$ is $\lambda$-strongly convex (which is not required), using the change of variable $\boldsymbol{w} := \boldsymbol{w}^t - \boldsymbol{d}$, we obtain

$$\nabla r_t^*(\boldsymbol{u}) = \underset{\boldsymbol{d} \in \mathbb{R}^p}{\operatorname{argmax}} \langle \boldsymbol{u}, \boldsymbol{d} \rangle - r_t(\boldsymbol{d})$$
$$= \underset{\boldsymbol{d} \in \mathbb{R}^p}{\operatorname{argmax}} \langle \boldsymbol{u}, \boldsymbol{d} \rangle - \frac{m}{2\gamma}\|\boldsymbol{d}\|_2^2 - m\,\rho(\boldsymbol{w}^t - \boldsymbol{d})$$
$$= \boldsymbol{w}^t - \underset{\boldsymbol{w} \in \mathbb{R}^p}{\operatorname{argmax}} -\langle \boldsymbol{u}, \boldsymbol{w} \rangle - \frac{m}{2\gamma}\|\boldsymbol{w}^t - \boldsymbol{w}\|_2^2 - m\,\rho(\boldsymbol{w})$$
$$= \boldsymbol{w}^t - \underset{\boldsymbol{w} \in \mathbb{R}^p}{\operatorname{argmin}} \|\boldsymbol{w} - (\boldsymbol{w}^t - \frac{\gamma}{m}\boldsymbol{u})\|_2^2 + \gamma\rho(\boldsymbol{w})$$
$$= \boldsymbol{w}^t - \operatorname{prox}_{\gamma\rho}\left(\boldsymbol{w}^t - \frac{\gamma}{m}\boldsymbol{u}\right).$$

**Sum of quadratic and $L_1$ regularizations.** As a particular example of the above, if $\rho(\boldsymbol{w}) = \lambda\|\boldsymbol{w}\|_1$, we obtain

$$\operatorname{prox}_{\gamma\rho}(\boldsymbol{z}) = \mathrm{ST}_{\lambda\gamma}(\boldsymbol{z}),$$

where we defined the soft-thresholding operator

$$\mathrm{ST}_\tau(\boldsymbol{z}) := \mathrm{prox}_{\tau\|\cdot\|_1}(\boldsymbol{z}) = \begin{cases} z_j - \tau, & z_j > \tau \\ 0, & |z_j| \le \tau \\ z_j + \tau, & z_j < -\tau \end{cases}.$$

## C   Proofs

### C.1   Derivation of the dual subproblem

We begin by deriving the dual when using a generic strongly convex regularizer $r_t$.

**Proposition 3.** *Denote $\boldsymbol{f}_i^t := f_i(\boldsymbol{w}^t)$, $J_i^t := \partial f_i(\boldsymbol{w}^t)$ and $r_t \colon \mathbb{R}^p \to \mathbb{R}$ a strongly convex regularizer. Then,*

$$\min_{\boldsymbol{d}\in\mathbb{R}^p} \sum_{i\in S} \ell_i(\boldsymbol{f}_i^t - J_i^t \boldsymbol{d}) + r_t(\boldsymbol{d}) = - \min_{\boldsymbol{\alpha}_S\in\mathbb{R}^{m\times k}} \sum_{i\in S} \left( \ell_i^*(\boldsymbol{\alpha}_i) - \langle \boldsymbol{f}_i^t, \boldsymbol{\alpha}_i \rangle \right) + r_t^* \left( (J_S^t)^* \boldsymbol{\alpha}_S \right)$$

*where $(J_S^t)^* \boldsymbol{\alpha}_S := \sum_{i=1}^m (J_i^t)^* \boldsymbol{\alpha}_i \in \mathbb{R}^p$, $\boldsymbol{\alpha}_S = (\boldsymbol{\alpha}_1, \dots, \boldsymbol{\alpha}_m)^\top \in \mathbb{R}^{m\times k}$.*

*The dual-primal link is*

$$\boldsymbol{d}^\star = \nabla r_t^* \left( (J_S^t)^* \boldsymbol{\alpha}_S^\star \right),$$

*where*

$$\nabla r_t^*(\boldsymbol{u}) = \underset{\boldsymbol{d}\in\mathbb{R}^p}{\mathrm{argmax}} \langle \boldsymbol{u}, \boldsymbol{d} \rangle - r_t(\boldsymbol{d}).$$

*Proof.*

$$\min_{\boldsymbol{d}\in\mathbb{R}^p} \sum_{i\in S} \ell_i(\boldsymbol{f}_i^t - J_i^t \boldsymbol{d}) + r_t(\boldsymbol{d})$$

$$= \min_{\boldsymbol{d}\in\mathbb{R}^p} \sum_{i=1}^m \max_{\boldsymbol{\alpha}_i\in\mathbb{R}^k} \langle \boldsymbol{f}_i^t - J_i^t \boldsymbol{d}, \boldsymbol{\alpha}_i \rangle - \ell_i^*(\boldsymbol{\alpha}_i) + r_t(\boldsymbol{d})$$

$$= \max_{\boldsymbol{\alpha}_1,\dots,\boldsymbol{\alpha}_m\in\mathbb{R}^k} \sum_{i\in S} \left( -\ell_i^*(\boldsymbol{\alpha}_i) + \langle \boldsymbol{f}_i^t, \boldsymbol{\alpha}_i \rangle \right) + \left[ \min_{\boldsymbol{d}\in\mathbb{R}^p} \left\langle -\sum_{i\in S}(J_i^t)^* \boldsymbol{\alpha}_i, \boldsymbol{d} \right\rangle + r_t(\boldsymbol{d}) \right]$$

$$= \max_{\boldsymbol{\alpha}_1,\dots,\boldsymbol{\alpha}_m\in\mathbb{R}^k} \sum_{i\in S} \left( -\ell_i^*(\boldsymbol{\alpha}_i) + \langle \boldsymbol{f}_i^t, \boldsymbol{\alpha}_i \rangle \right) - \left[ \max_{\boldsymbol{d}\in\mathbb{R}^p} \left\langle \sum_{i\in S}(J_i^t)^* \boldsymbol{\alpha}_i, \boldsymbol{d} \right\rangle - r_t(\boldsymbol{d}) \right]$$

$$= \max_{\boldsymbol{\alpha}_1,\dots,\boldsymbol{\alpha}_m\in\mathbb{R}^k} \sum_{i\in S} \left( -\ell_i^*(\boldsymbol{\alpha}_i) + \langle \boldsymbol{f}_i^t, \boldsymbol{\alpha}_i \rangle \right) - r_t^* \left( \sum_{i\in S}(J_i^t)^* \boldsymbol{\alpha}_i \right).$$

$\square$

### C.2   Approximate dual subproblem

Consider the dual subproblem derived in Proposition 3, that is,

$$\min_{\boldsymbol{\alpha}_S\in\mathbb{R}^{m\times k}} \ell_S^*(\boldsymbol{\alpha}_S) - \langle \boldsymbol{f}_S^t, \boldsymbol{\alpha}_S \rangle + r_t^* \left( (J_S^t)^* \boldsymbol{\alpha}_S \right).$$

Unfortunately, this subproblem could be difficult to solve for generic $r_t$. In our case $r_t$ takes the form $r_t(\boldsymbol{d}) = m \left[ \rho(\boldsymbol{w}^t - \boldsymbol{d}) + \frac{1}{2\gamma}\|\boldsymbol{d}\|_2^2 \right]$, which is strongly convex. We can therefore exploit the smoothness of its convex conjugate $r_t^*$. Inspired by the prox-SDCA algorithm (Shalev-Shwartz & Zhang, 2013), we therefore propose to approximate $r_t^*$ by a quadratic upper bound.

**Proposition 4.** *If $r_t$ is $\mu$-strongly convex, the solution of the dual subproblem*

$$\min_{\boldsymbol{\alpha}_S\in\mathbb{R}^{m\times k}} \ell_S^*(\boldsymbol{\alpha}_S) - \langle \boldsymbol{f}_S^t, \boldsymbol{\alpha}_S \rangle + r_t^* \left( (J_S^t)^* \boldsymbol{\alpha}_S \right), \tag{13}$$

*can be approximated around any $\boldsymbol{u}^t \in \mathbb{R}^p$ by solving*

$$\min_{\boldsymbol{\alpha}_S \in \mathbb{R}^{m \times k}} \ell_S^*(\boldsymbol{\alpha}_S) - \langle \boldsymbol{f}_S^t - J_S^t \left( \nabla r_t^*(\boldsymbol{u}^t) - \frac{1}{\mu} \boldsymbol{u}^t \right), \boldsymbol{\alpha}_S \rangle + \frac{1}{2\mu} \|(J_S^t)^* \boldsymbol{\alpha}_S\|^2.$$

*Proof.* Denoting $\| \cdot \|$ the norm w.r.t. which $r_t^*$ is $\frac{1}{\mu}$-smooth, we have for any $\boldsymbol{u}, \boldsymbol{v} \in \mathbb{R}^p$,

$$r_t^*(\boldsymbol{v}) \le \boldsymbol{r}_t(\boldsymbol{u}) + \langle \nabla r_t^*(\boldsymbol{u}), \boldsymbol{v} - \boldsymbol{u} \rangle + \frac{1}{2\mu} \|\boldsymbol{u} - \boldsymbol{v}\|^2. \tag{14}$$

Using $\boldsymbol{v}^t = (J_S^t)^* \boldsymbol{\alpha}_S$, we obtain

$$r_t^* \left( (J_S^t)^* \boldsymbol{\alpha}_S \right) \le r_t(\boldsymbol{u}^t) + \langle \nabla r_t^*(\boldsymbol{u}^t), (J_S^t)^* \boldsymbol{\alpha}_S - \boldsymbol{u}^t \rangle + \frac{1}{2\mu} \|(J_S^t)^* \boldsymbol{\alpha}_S - \boldsymbol{u}^t\|_2^2$$

$$= \langle J_S^t \nabla r_t^*(\boldsymbol{u}^t), \boldsymbol{\alpha}_S \rangle + \frac{1}{2\mu} \|(J_S^t)^* \boldsymbol{\alpha}_S\|_2^2 - \frac{1}{\mu} \langle J_S^t \boldsymbol{u}, \boldsymbol{\alpha}_S \rangle + \text{const w.r.t. } \boldsymbol{\alpha}_S.$$

Plugging this quadratic upper-bound of $r_t^* \left( (J_S^t)^* \boldsymbol{\alpha}_S \right)$ back in (13), we arrive at the approximate subproblem

$$\min_{\boldsymbol{\alpha}_S \in \mathbb{R}^{m \times k}} \ell_S^*(\boldsymbol{\alpha}_S) - \left\langle \boldsymbol{f}_S^t - J_S^t \left( \nabla r_t^*(\boldsymbol{u}^t) - \frac{1}{\mu} \boldsymbol{u}^t \right), \boldsymbol{\alpha}_S \right\rangle + \frac{1}{2\mu} \|(J_S^t)^* \boldsymbol{\alpha}_S\|^2.$$

$\square$

## C.3 Dual of quadratic approximation of convex losses

When using a quadratic-linear approximation of $h_i$, the primal subproblem is, for a mini-batch $S = \{i_1, \ldots, i_m\} \subseteq [n]$,

$$\boldsymbol{d}_S^t \coloneqq \operatorname*{argmin}_{\boldsymbol{d} \in \mathbb{R}^p} q_S^t(-J_i^t \boldsymbol{d}) - \langle \boldsymbol{f}_S^t, \boldsymbol{d} \rangle + \frac{m}{2\gamma} \|\boldsymbol{d}\|_2^2$$

$$= \operatorname*{argmin}_{\boldsymbol{d} \in \mathbb{R}^p} \sum_{i \in S} \frac{1}{2} \langle \boldsymbol{d}, (J_i^t)^* H_i^t J_i^t \boldsymbol{d} \rangle - \langle \boldsymbol{d}, (J_i^t)^* \boldsymbol{g}_i^t \rangle + \frac{m}{2\gamma} \|\boldsymbol{d}\|_2^2$$

$$= \operatorname*{argmin}_{\boldsymbol{d} \in \mathbb{R}^p} \frac{1}{2} \langle \boldsymbol{d} (J_S^t)^* H_S^t J_S^t \boldsymbol{d} \rangle - \langle \boldsymbol{d}, (J_S^t)^* \boldsymbol{g}_S^t \rangle + \frac{m}{2\gamma} \|\boldsymbol{d}\|_2^2,$$

$$= \operatorname*{argmin}_{\boldsymbol{d} \in \mathbb{R}^p} a_S(-J_S^t \boldsymbol{d}) + \frac{m}{2\gamma} \|\boldsymbol{d}\|_2^2,$$

where we used the shorthands

$$a_S^t(\boldsymbol{z}) \coloneqq q_S^t(\boldsymbol{z}) - \langle \boldsymbol{f}_S^t, \boldsymbol{z} \rangle = \frac{1}{2} \langle \boldsymbol{z}, H_S^t \boldsymbol{z} \rangle + \langle \boldsymbol{z}, \boldsymbol{g}_S^t \rangle = b_S^t(\boldsymbol{z}) + \langle \boldsymbol{z}, \boldsymbol{g}_S^t \rangle$$

$$b_S^t(\boldsymbol{z}) \coloneqq \sum_{j=1}^m b_{i_j}^t(\boldsymbol{z}_j) = \sum_{j=1}^m \frac{1}{2} \langle \boldsymbol{z}_j, H_{i_j}^t \boldsymbol{z}_j \rangle = \frac{1}{2} \langle \boldsymbol{z}, H_S^t \boldsymbol{z} \rangle,$$

$$b_i^t(\boldsymbol{z}) \coloneqq \frac{1}{2} \langle \boldsymbol{z}, H_i^t \boldsymbol{z} \rangle.$$

We recall that

$$\boldsymbol{f}_i^t \coloneqq f_i(\boldsymbol{w}^t)$$
$$J_i^t \coloneqq \partial f_i(\boldsymbol{w}^t)$$
$$\boldsymbol{g}_i^t \coloneqq \nabla \ell_i(\boldsymbol{f}_i^t)$$
$$H_i^t \coloneqq \nabla^2 \ell_i(\boldsymbol{f}_i^t)$$

and similarly

$$
\begin{aligned}
\boldsymbol{f}_S^t &:= f_S(\boldsymbol{w}^t) := (f_{i_1}(\boldsymbol{w}^t), \ldots, f_{i_m}(\boldsymbol{w}^t)) \\
\boldsymbol{g}_S^t &:= \nabla \ell_S(\boldsymbol{f}_S^t) \\
H_S^t \boldsymbol{u} &:= \nabla^2 \ell_S(\boldsymbol{f}_S^t) \boldsymbol{u} = (\nabla^2 \ell_{i_1}(\boldsymbol{f}_{i_1}^t) \boldsymbol{u}_1, \ldots, \nabla^2 \ell_{i_m}(\boldsymbol{f}_{i_m}^t) \boldsymbol{u}_m) \\
J_S^t \boldsymbol{d} &:= \partial f_S(\boldsymbol{w}^t) \boldsymbol{d} := (\partial f_{i_1}(\boldsymbol{w}^t) \boldsymbol{d}, \ldots, \partial f_{i_m}(\boldsymbol{w}^t) \boldsymbol{d}) \\
(J_S^t)^* \boldsymbol{u} &:= \partial f_S(\boldsymbol{w}^t)^* \boldsymbol{u} = \sum_{j=1}^m \partial f_{i_j}(\boldsymbol{w}^t)^* \boldsymbol{u}_j.
\end{aligned}
$$

**Strictly convex losses.** We first present the result for strictly convex losses, in which case the convex conjugate of interest is well-known.

**Proposition 5.** *The prox-linear direction associated to a linear quadratic approximation of the objective*

$$
\boldsymbol{d}_S^t := \operatorname*{argmin}_{\boldsymbol{d} \in \mathbb{R}^p} \frac{1}{2} \langle \boldsymbol{d}, (J_S^t)^* H_S^t J_S^t \boldsymbol{d} \rangle - \langle \boldsymbol{d}, (J_S^t)^* \boldsymbol{g}_S^t \rangle + \frac{m}{2\gamma} \|\boldsymbol{d}\|_2^2,
$$

*with $H_i^t$ and so $H_S^t$ invertible, can be computed as $\boldsymbol{d}_S^t = \frac{\gamma}{m}(J_S^t)^* \boldsymbol{\alpha}_S^t$, $\boldsymbol{\alpha}_S^t = \boldsymbol{g}_S^t - \boldsymbol{\beta}_S^t$ for*

$$
\boldsymbol{\beta}_S^t := \operatorname*{argmin}_{\boldsymbol{\beta} \in \mathbb{R}^{m \times k}} \frac{1}{2} \langle \boldsymbol{\beta}, (H_S^t)^{-1} \boldsymbol{\beta} \rangle + \frac{\gamma}{2m} \|(J_S^t)^*(\boldsymbol{g}_S^t - \boldsymbol{\beta})\|_2^2,
$$

*where*

$$
(H_S^t)^{-1} \boldsymbol{\beta} := ((H_{i_1}^t)^{-1} \boldsymbol{\beta}_1, \ldots, (H_{i_m}^t)^{-1} \boldsymbol{\beta}_m).
$$

*Proof.* We use the same notations as in the beginning of the section. The convex conjugate of $a_S^t$ can be expressed in terms of the convex conjugate of $b_S^t$ as

$$
(a_S^t)^*(\boldsymbol{\alpha}) = (b_S^t)^*(\boldsymbol{\alpha} - \boldsymbol{g}_S^t).
$$

The convex conjugate of $b_S^t$ itself can be expressed as

$$
(b_S^t)^*(\boldsymbol{\beta}) = \sum_{j=1}^n (b_{i_j}^t)^*(\boldsymbol{\beta}_j) = \sum_{j=1}^m \frac{1}{2} \langle \boldsymbol{\beta}_j, (H_{i_j}^t)^{-1} \boldsymbol{\beta}_j \rangle = \frac{1}{2} \langle \boldsymbol{\beta}, (H_S^t)^{-1} \boldsymbol{\beta} \rangle,
$$

using that $H_i^t$ is invertible such that $(b_i^t)^*(\boldsymbol{\beta}) = \frac{1}{2} \langle \boldsymbol{\beta}, (H_i^t)^{-1} \boldsymbol{\beta} \rangle$. The problem can then be solved as $\boldsymbol{d}_S^t = (J_S^t)^* \boldsymbol{\alpha}_S^t$ for

$$
\begin{aligned}
\boldsymbol{\alpha}_S^t &:= \operatorname*{argmin}_{\boldsymbol{\alpha} \in \mathbb{R}^{m \times k}} (a_S^t)^*(\boldsymbol{\alpha}) + \frac{\gamma}{2m} \|(J_S^t)^* \boldsymbol{\alpha}\|_2^2 = \operatorname*{argmin}_{\boldsymbol{\alpha} \in \mathbb{R}^{m \times k}} (b_S^t)^*(\boldsymbol{\alpha} - \boldsymbol{g}_S^t) + \frac{\gamma}{2m} \|(J_S^t)^* \boldsymbol{\alpha}\|_2^2 = \boldsymbol{g}_S^t - \boldsymbol{\beta}_S^t \\
\boldsymbol{\beta}_S^t &:= \operatorname*{argmin}_{\boldsymbol{\beta} \in \mathbb{R}^{m \times k}} (b_S^t)^*(-\boldsymbol{\beta}) + \frac{\gamma}{2m} \|(J_S^t)^*(\boldsymbol{g}_S^t - \boldsymbol{\beta})\|_2^2 = \operatorname*{argmin}_{\boldsymbol{\beta} \in \mathbb{R}^{m \times k}} \frac{1}{2} \langle \boldsymbol{\beta}, (H_S^t)^{-1} \boldsymbol{\beta} \rangle + \frac{\gamma}{2m} \|(J_S^t)^*(\boldsymbol{g}_S^t - \boldsymbol{\beta})\|_2^2.
\end{aligned}
$$

$\square$

The approach above holds for example in the case of the squared loss. The logistic loss on the other hand is not strictly convex, therefore its Hessian is not invertible. We present below a generic derivation for any convex loss. We then specialize the result for the logistic loss.

**Generic convex loss.** In the generic case, we can tackle the computation of the dual of the quadratic-linear approximation by using the pseudo-inverse of the Hessian as stated in Proposition 6.

**Proposition 6.** *The prox-linear direction associated to a linear quadratic approximation of the objective*

$$\boldsymbol{d}_S^t = \underset{\boldsymbol{d} \in \mathbb{R}^p}{\mathrm{argmin}}\, \frac{1}{2}\langle \boldsymbol{d}, (J_S^t)^* H_S^t J_S^t \boldsymbol{d}\rangle - \langle \boldsymbol{d}, (J_S^t)^* \boldsymbol{g}_S^t\rangle + \frac{m}{2\gamma}\|\boldsymbol{d}\|_2^2,$$

*can be computed as $\boldsymbol{d}_S^t = \frac{\gamma}{m}(J_S^t)^* \boldsymbol{\alpha}_S^t$, $\boldsymbol{\alpha}_S^t = \boldsymbol{g}_S^t - \boldsymbol{\beta}_S^t$ for*

$$\boldsymbol{\beta}_S^t = \underset{\boldsymbol{\beta} \in \mathbb{R}^{m \times k}}{\mathrm{argmin}}\, \frac{1}{2}\langle \boldsymbol{\beta}, (H_S^t)^\dagger \boldsymbol{\beta}\rangle + \frac{\gamma}{2m}\|(J_S^t)^*(\boldsymbol{g}_S^t - \boldsymbol{\beta})\|_2^2$$

$$s.t. \quad (I - H_S^t (H_S^t)^\dagger)\boldsymbol{\beta} = \boldsymbol{0},$$

*where $(H_i^t)^\dagger$ denotes the pseudo inverse of $H_i^t$ and*

$$(H_S^t)^\dagger \boldsymbol{\beta} = ((H_{i_1}^t)^\dagger \boldsymbol{\beta}_1, \ldots, (H_{i_m}^t)^\dagger \boldsymbol{\beta}_m).$$

*Proof.* The proof follows the same reasoning as in Proposition 5 except that for generic convex loss, the convex conjugate of $b_i^t$ is given by Lemma 7 as

$$(b_i^t)^*(\boldsymbol{\beta}) = \begin{cases} \frac{1}{2}\langle \boldsymbol{\beta}, (H_i^t)^\dagger \boldsymbol{\beta}\rangle & \text{if } (I - H_i^t (H_i^t)^\dagger)\boldsymbol{\beta} = 0 \\ +\infty & \text{otherwise.} \end{cases}$$

The result follows using that $H_S^t (H_S^t)^\dagger \boldsymbol{\beta} = (H_{i_1}^t (H_{i_1}^t)^\dagger \boldsymbol{\beta}_1, \ldots, H_{i_m}^t (H_{i_m}^t)^\dagger \boldsymbol{\beta}_m)$. $\qquad\square$

**Lemma 7.** *Let $q(\boldsymbol{w}) := \frac{1}{2}\langle \boldsymbol{w}, A\boldsymbol{w}\rangle$, where $A \succeq 0$, $A \in \mathbb{R}^{k \times k}$. The convex conjugate of $q$ is*

$$q^*(\boldsymbol{v}) = \begin{cases} \frac{1}{2}\langle \boldsymbol{v}, A^\dagger \boldsymbol{v}\rangle & \text{if } AA^\dagger \boldsymbol{v} = \boldsymbol{v} \\ +\infty & \text{otherwise} \end{cases}$$

*where $A^\dagger$ denotes the pseudo-inverse of $A$.*

*Proof.* Denote $P = I - A^\dagger A$ the projection on the null-space of $A$. Note that as $A$ is symmetric, we have $P = I - AA^\dagger$. Since $A \succeq 0$, $q$ is convex and its conjugate is defined as

$$q^*(\boldsymbol{v}) = \sup_{\boldsymbol{w} \in \mathbb{R}^k} \langle \boldsymbol{v}, \boldsymbol{w}\rangle - \frac{1}{2}\langle \boldsymbol{w}, A\boldsymbol{w}\rangle.$$

If $P\boldsymbol{v} \neq 0$, then by considering $\boldsymbol{w}(t) = tP\boldsymbol{v}$ for $t \in \mathbb{R}$, we have $\langle \boldsymbol{v}, \boldsymbol{w}(t)\rangle - \frac{1}{2}\langle \boldsymbol{w}(t), A\boldsymbol{w}(t)\rangle = t\|P\boldsymbol{v}\|_2^2$ which tends to $+\infty$ for $t \to +\infty$. Hence, $q^*(\boldsymbol{v}) = +\infty$ if $P\boldsymbol{v} \neq 0$.

If $\boldsymbol{v} = AA^\dagger \boldsymbol{v}$. The convex conjugate then amounts to solve

$$\sup_{\boldsymbol{w} \in \mathbb{R}^d} \langle \boldsymbol{w}, AA^\dagger \boldsymbol{v}\rangle - \frac{1}{2}\langle \boldsymbol{w}, A\boldsymbol{w}\rangle.$$

The solution of this problem is given by $\boldsymbol{w}^\star$ such that $AA^\dagger \boldsymbol{v} = A\boldsymbol{w}^\star$, hence $\boldsymbol{w}^\star = A^\dagger \boldsymbol{v}$ is a solution and the convex conjugate is then

$$q^*(\boldsymbol{v}) = \langle \boldsymbol{v}, A^{\dagger^*} AA^\dagger \boldsymbol{v}\rangle - \frac{1}{2}\langle \boldsymbol{v}, A^{\dagger^*} AA^\dagger \boldsymbol{v}\rangle = \langle \boldsymbol{v}, A^\dagger AA^\dagger \boldsymbol{v}\rangle - \frac{1}{2}\langle \boldsymbol{v}, A^\dagger AA^\dagger \boldsymbol{v}\rangle = \frac{1}{2}\langle \boldsymbol{v}, A^\dagger \boldsymbol{v}\rangle,$$

where we used that $A^\dagger$ is symmetric since $A$ is symmetric, and we used the identity $A^\dagger AA^\dagger = A^\dagger$. $\qquad\square$

**Logistic loss.** We now derive the conjugate of the quadratic approximation in the case of the logistic loss.

**Proposition 8.** *Consider the logistic loss*

$$\ell(\boldsymbol{f}) = -\langle \boldsymbol{y}, \boldsymbol{f} \rangle + \phi(\boldsymbol{f}),$$

*for $\boldsymbol{y} \in \{0,1\}^k$, $\boldsymbol{y}^\top \mathbf{1}_k = 1$ and $\phi(\boldsymbol{f}) = \log(\exp(\boldsymbol{f})^\top \mathbf{1}_k)$ for $\boldsymbol{f} \in \mathbb{R}^k$, where $\exp$ is applied element-wise. Consider the quadratic approximation of the logistic loss at a point $\boldsymbol{f}$ given by*

$$q(\ell, \boldsymbol{f})(\boldsymbol{v}) := -\langle \boldsymbol{y} - \nabla\phi(\boldsymbol{f}), \boldsymbol{v} \rangle + \frac{1}{2}\boldsymbol{v}^\top \nabla^2\phi(\boldsymbol{f})\boldsymbol{v},$$

*where $\nabla\phi(\boldsymbol{f}) = \sigma(\boldsymbol{f}), \nabla^2\phi(\boldsymbol{f}) = \mathbf{diag}\left(\sigma(\boldsymbol{f})\right) - \sigma(\boldsymbol{f})\sigma(\boldsymbol{f})^\top, \sigma(\boldsymbol{f}) := \mathrm{softmax}(\boldsymbol{f}) = \exp(\boldsymbol{f})/(\exp(\boldsymbol{f})^\top \mathbf{1}_k)$. Its convex conjugate is, for $\boldsymbol{\beta} := \boldsymbol{\alpha} - \nabla\ell(\boldsymbol{f})$ and $D := \mathbf{diag}(\sigma(\boldsymbol{f}))$,*

$$q(\ell, \boldsymbol{f})^*(\boldsymbol{\alpha}) = \begin{cases} \frac{1}{2}\langle \boldsymbol{\beta}, D^{-1}\boldsymbol{\beta} \rangle & \text{if } \boldsymbol{\beta}^\top \mathbf{1}_k = 0 \\ +\infty & \text{otherwise.} \end{cases}$$

*Proof.* The convex conjugate reads

$$q(\ell, \boldsymbol{f})^*(\boldsymbol{\alpha}) = h^*(\boldsymbol{\alpha} + \boldsymbol{y} - \nabla\phi(\boldsymbol{f})),$$

where

$$h^*(\boldsymbol{\beta}) = \sup_{\boldsymbol{v} \in \mathbb{R}^k} \boldsymbol{\beta}^\top \boldsymbol{v} - \frac{1}{2}\langle \boldsymbol{v}, \nabla^2\phi(\boldsymbol{f})\boldsymbol{v} \rangle.$$

Note that $\nabla^2\phi(\boldsymbol{f})\mathbf{1}_k = 0$. In the following, denote $\boldsymbol{\beta} = \boldsymbol{\alpha} + \boldsymbol{y} - \nabla\phi(\boldsymbol{f})$ and consider computing $h^*(\boldsymbol{\beta})$. Note that $\boldsymbol{\beta}^\top \mathbf{1}_k = \boldsymbol{\alpha}^\top \mathbf{1}_k$ since $\boldsymbol{y}^\top \mathbf{1}_k = \nabla\phi(\boldsymbol{f})^\top \mathbf{1}_k = 1$.

If $\boldsymbol{\beta}^\top \mathbf{1}_k \neq 0$, that is $\boldsymbol{\alpha}^\top \mathbf{1}_k \neq 0$, then by considering $\boldsymbol{v}(t) = t\,\mathbf{1}_k \mathbf{1}_k^\top \boldsymbol{\beta}$, we have

$$\boldsymbol{\beta}^\top \boldsymbol{v}(t) - \frac{1}{2}\langle \boldsymbol{v}(t), \nabla^2\phi(\boldsymbol{f})\boldsymbol{v}(t) \rangle = t(\mathbf{1}_k^\top \boldsymbol{\beta})^2 \xrightarrow[t \to +\infty]{} +\infty$$

so $h^*(\boldsymbol{\beta}) = +\infty$ and $q(\ell, \boldsymbol{f})^*(\boldsymbol{\alpha}) = +\infty$.

Consider now $\boldsymbol{\beta}^\top \mathbf{1}_k = 0$, that is $\boldsymbol{\alpha}^\top \mathbf{1}_k = 0$ and $\boldsymbol{v}^\star = D^{-1}\boldsymbol{\beta}$ for $D = \mathbf{diag}\left(\frac{\exp(\boldsymbol{f})}{\exp(\boldsymbol{f})^\top \mathbf{1}_k}\right)$. We have then

$$\nabla^2\phi(\boldsymbol{f})\boldsymbol{v}^\star = DD^{-1}\boldsymbol{\beta} - \frac{\exp(\boldsymbol{f})}{\exp(\boldsymbol{f})^\top \mathbf{1}_k}\mathbf{1}_k^\top \boldsymbol{\beta} = \boldsymbol{\beta}.$$

Hence, $\boldsymbol{v}^\star$ satisfies the first-order conditions of the problem defining $h^*(\boldsymbol{\beta})$, so it is a solution of that problem. The expression of the convex conjugate follows. □

Using this expression of the conjugate, we obtain (10) as formally stated below.

**Corollary 9.** *The prox-linear direction associated to a linear quadratic approximation of the objective*

$$\boldsymbol{d}_S^t = \underset{\boldsymbol{d} \in \mathbb{R}^p}{\operatorname{argmin}} \frac{1}{2}\langle \boldsymbol{d}, (J_S^t)^* H_S^t J_S^t \boldsymbol{d} \rangle - \langle \boldsymbol{d}, (J_S^t)^* \boldsymbol{g}_i^t \rangle + \frac{m}{2\gamma}\|\boldsymbol{d}\|_2^2,$$

*for $\ell_i$ the logistic loss, can be computed as $\boldsymbol{d}_S^t = \frac{\gamma}{m}(J_S^t)^* \boldsymbol{\alpha}_S^t$, $\boldsymbol{\alpha}_S^t = \boldsymbol{g}_S^t - \boldsymbol{\beta}_S^t$ for*

$$\boldsymbol{\beta}_S^t = \underset{\boldsymbol{\beta} \in \mathbb{R}^{m \times k}}{\operatorname{argmin}} \frac{1}{2}\langle \boldsymbol{\beta}, (D_S^t)^{-1}\boldsymbol{\beta} \rangle + \frac{\gamma}{2m}\|(J_S^t)^*(\boldsymbol{g}_S^t - \boldsymbol{\beta})\|_2^2,$$

$$\text{s.t.} \quad \mathbf{1}_k^\top \boldsymbol{\beta}_i = \mathbf{0} \text{ for } i \in \{1, \ldots m\}$$

*for $(D_S^t)^{-1}\boldsymbol{\beta} = (\boldsymbol{\beta}_1/\sigma(\boldsymbol{f}_{i_1}^t), \ldots, \boldsymbol{\beta}_m/\sigma(\boldsymbol{f}_{i_m}^t))$.*

### C.4 Conjugate gradient method for quadratic approximations

**Proposition 10.** *Consider a quadratic problem under linear constraints of the form*

$$\min_{\boldsymbol{\beta}\in\mathbb{R}^d} \frac{1}{2}\langle\boldsymbol{\beta}, Q\boldsymbol{\beta}\rangle - \langle\boldsymbol{\beta}, \boldsymbol{c}\rangle \tag{15}$$
$$s.t.\ (I-P)\boldsymbol{\beta} = 0$$

*for $Q$ semi-definite positive and $P$ an orthonormal projector, that is, $P = P^*$ and $PP = P$. Assume that $\boldsymbol{\beta}\mapsto\frac{1}{2}\langle\boldsymbol{\beta}, Q\boldsymbol{\beta}\rangle - \langle\boldsymbol{\beta}, \boldsymbol{c}\rangle$ is bounded below.*

*Any convergent first-order optimization algorithm applied to the unconstrained problem*

$$\min_{\boldsymbol{\beta}\in\mathbb{R}^d} \frac{1}{2}\langle\boldsymbol{\beta}, PQP\boldsymbol{\beta}\rangle - \langle\boldsymbol{\beta}, P\boldsymbol{c}\rangle \tag{16}$$

*and initialized at $\boldsymbol{\beta}^0 = \boldsymbol{0}$ converges to a solution of (15).*

*Proof.* First, note that problem (15) is necessarily feasible as its constraints are satisfied for $\boldsymbol{\beta} = \boldsymbol{0}$. Moreover, problem (15) admits a minimizer since it is the minimization of a convex quadratic bounded below on a subspace of $\mathbb{R}^d$.

Similarly, if $\boldsymbol{\beta}\mapsto\frac{1}{2}\langle\boldsymbol{\beta}, Q\boldsymbol{\beta}\rangle - \langle\boldsymbol{\beta}, \boldsymbol{c}\rangle$ is bounded below, then necessarily $\boldsymbol{c}$ does not belong to the null space of $Q$, otherwise taking $\boldsymbol{\beta} = t\boldsymbol{c}$, $t\to+\infty$, would lead to $-\infty$. Since the null space and the image of $Q$ are orthonormal spaces, $\boldsymbol{c}$ belongs to the image of $Q$, so that there exists $\boldsymbol{d}\in\mathbb{R}^d$ satisfying $\boldsymbol{c} = Q\boldsymbol{d}$. The objective of (16) can then be factorized as $\frac{1}{2}\langle\boldsymbol{\beta}, PQP\boldsymbol{\beta}\rangle - \langle\boldsymbol{\beta}, P\boldsymbol{c}\rangle = \frac{1}{2}\langle(P\boldsymbol{\beta}-\boldsymbol{d}), Q(P\boldsymbol{\beta}-\boldsymbol{d})\rangle - \frac{1}{2}\langle\boldsymbol{d}, Q\boldsymbol{d}\rangle$. Hence, it is a convex quadratic bounded below, so it admits a minimizer.

A point $\boldsymbol{\beta}^\star$ is optimal for (15) if there exists $\boldsymbol{\lambda}^\star\in\mathbb{R}^d$ such that

$$PQP\boldsymbol{\beta}^\star - P\boldsymbol{c} + P\boldsymbol{\lambda}^\star - \boldsymbol{\lambda}^\star = \boldsymbol{0}, \quad P\boldsymbol{\beta}^\star = \boldsymbol{\beta}^\star.$$

In comparison, a convergent first-order optimization algorithm applied to (16) converges to a point $\hat{\boldsymbol{\beta}}$ satisfying the first order optimality conditions of (16), that is,

$$PQP\hat{\boldsymbol{\beta}} - P\boldsymbol{c} = \boldsymbol{0}.$$

The iterates of any first-order optimization algorithm are built such that

$$\boldsymbol{\beta}^k\in\mathrm{Span}(\boldsymbol{\beta}^0, \nabla f(\boldsymbol{\beta}^0), \boldsymbol{\beta}^1, \nabla f(\boldsymbol{\beta}^1), \ldots, \boldsymbol{\beta}^{k-1}, \nabla f(\boldsymbol{\beta}^{k-1})),$$

where $f(\boldsymbol{\beta}) = \frac{1}{2}\langle\boldsymbol{\beta}, PQP\boldsymbol{\beta}\rangle - \langle\boldsymbol{\beta}, P\boldsymbol{c}\rangle$. Denote $\mathcal{C}\coloneqq\{\boldsymbol{\beta} : P\boldsymbol{\beta} = \boldsymbol{\beta}\}$, that is a subspace of $\mathbb{R}^d$. We have that for any $\boldsymbol{\beta}\in\mathbb{R}^d$, $\nabla f(\boldsymbol{\beta}) = PQP\boldsymbol{\beta} - P\boldsymbol{c}\in\mathcal{C}$ since $PP = P$. If $\boldsymbol{\beta}^0 = \boldsymbol{0}$, then $\boldsymbol{\beta}^0\in\mathcal{C}$ and by induction we have that $\boldsymbol{\beta}^k\in\mathcal{C}$. Therefore, $\hat{\boldsymbol{\beta}} = \lim_{k\to+\infty}\boldsymbol{\beta}^k$ satisfies $P\hat{\boldsymbol{\beta}} = \lim_{k\to+\infty}P\boldsymbol{\beta}^k = \lim_{k\to+\infty}\boldsymbol{\beta}^k = \hat{\boldsymbol{\beta}}$. Therefore, $\hat{\boldsymbol{\beta}}$ satisfies

$$PQP\hat{\boldsymbol{\beta}} - P\boldsymbol{c} = 0, \quad P\hat{\boldsymbol{\beta}} = \hat{\boldsymbol{\beta}}.$$

It is therefore a solution of (15) with associated $\boldsymbol{\lambda}^\star = \boldsymbol{0}$. □

The above proposition can directly be applied to the problems associated to $\boldsymbol{\beta}_S^t$ in Proposition 6 and 8 they are convex quadratic problem unbounded below.

**Corollary 11.** *The prox-linear direction associated to a linear quadratic approximation of the objective*

$$\boldsymbol{d}_S^t = \underset{\boldsymbol{d}\in\mathbb{R}^p}{\mathrm{argmin}}\ \frac{1}{2}\langle\boldsymbol{d}, (J_S^t)^* H_S^t J_S^t\boldsymbol{d}\rangle - \langle\boldsymbol{d}, (J_S^t)^* \boldsymbol{g}_S^t\rangle + \frac{m}{2\gamma}\|\boldsymbol{d}\|_2^2,$$

*can be computed as $\boldsymbol{d}_S^t = \frac{\gamma}{m}(J_S^t)^*\boldsymbol{\alpha}_S^t$, $\boldsymbol{\alpha}_S^t = \boldsymbol{g}_S^t - \boldsymbol{\beta}_S^t$ for $\boldsymbol{\beta}_S^t$ solving*

$$\boldsymbol{\beta}_S^t = \underset{\boldsymbol{\beta}\in\mathbb{R}^{m\times k}}{\mathrm{argmin}}\ \frac{1}{2}\langle\boldsymbol{\beta}, P(H_S^t)^\dagger P\boldsymbol{\beta}\rangle + \frac{\gamma}{2m}\|(J_S^t)^*(\boldsymbol{g}_S^t - P\boldsymbol{\beta})\|_2^2,$$

*for $P = H_S^t(H_S^t)^\dagger$ and $\boldsymbol{\beta}_S^t$ computed by a conjugate gradient method initialized at $\boldsymbol{0}$.*

**Corollary 12.** *The prox-linear direction associated to a linear quadratic approximation of the objective*

$$\boldsymbol{d}_S^t = \operatorname*{argmin}_{\boldsymbol{d} \in \mathbb{R}^p} \frac{1}{2} \langle \boldsymbol{d}, (J_S^t)^* H_S^t J_S^t \boldsymbol{d} \rangle - \langle \boldsymbol{d}, (J_S^t)^* \boldsymbol{g}_S^t \rangle + \frac{m}{2\gamma} \|\boldsymbol{d}\|_2^2,$$

*for $\ell_i$ the logistic loss, can be computed as $\boldsymbol{d}_S^t = \frac{\gamma}{m}(J_S^t)^* \boldsymbol{\alpha}_S^t$, $\boldsymbol{\alpha}_S^t = \boldsymbol{g}_S^t - \boldsymbol{\beta}_S^t$ for $\boldsymbol{\beta}_S^t$ solving*

$$\boldsymbol{\beta}_S^t = \operatorname*{argmin}_{\boldsymbol{\beta} \in \mathbb{R}^{m \times k}} \frac{1}{2} \langle \boldsymbol{\beta}, P(D_S^t)^{-1} P \boldsymbol{\beta} \rangle + \frac{\gamma}{2m} \|(J_S^t)^*(\boldsymbol{g}_S^t - P\boldsymbol{\beta})\|_2^2,$$

*for $P\boldsymbol{\beta} = (\Pi_k \boldsymbol{\beta}_1, \ldots, \Pi_k \boldsymbol{\beta}_m)$, $\Pi_k = \frac{1}{n} \mathbf{1}_k \mathbf{1}_k^\top$ and $\boldsymbol{\beta}_S^t$ computed by a conjugate gradient method initialized at $\mathbf{0}$.*

In practice, the matrix $D^{-1}$ may also be ill-conditioned as it consists of the diagonal of the reciprocal of the softmax, whose values may be close to 0. To avoid this problem, we precondition the problem associated with $\boldsymbol{\beta}_S^t$ by $D^{1/2}$.

## C.5 Prox-linear directions define critical points

We recall here that prox-linear directions define critical points. This will help us refine the results about descent directions. We state it for individual functions for simplicity. The result can readily be generalized for mini-batches. Nullity of $\boldsymbol{d}(\gamma \ell_i, f_i)(\boldsymbol{w}^t)$ can also be linked to the fact that $\boldsymbol{w}^t$ is close to a stationary point as shown by Drusvyatskiy & Paquette (2019).

**Proposition 13.** *If $\boldsymbol{w}^\star$ is a minimum of $\ell_i \circ f_i$ then the prox-linear direction (2) $\boldsymbol{d}(\gamma \ell_i, f_i)(\boldsymbol{w}^\star)$ or its quadratic approximation (4) $\boldsymbol{d}(\gamma q_i^\star, f_i)(\boldsymbol{w}^\star)$, with $q_i^\star$ the quadratic approximation of $\ell_i$ around $f_i(\boldsymbol{w}^\star)$, is zero. On the other hand, if $\boldsymbol{d}(\gamma q_i^\star, f_i)(\boldsymbol{w}^\star) = \mathbf{0}$, then $\nabla(\ell_i \circ f_i)(\boldsymbol{w}^\star) = \mathbf{0}$, that is $\boldsymbol{w}^\star$ is a critical point.*

*Proof.* Suppose $\boldsymbol{w}^\star$ is a minimum of $\ell_i \circ f_i$. Denote $F(\boldsymbol{d}) = \ell_i(f_i(\boldsymbol{w}^\star) - \partial f_i(\boldsymbol{d})) + \gamma \|\boldsymbol{d}\|_2^2/2$. Since $\boldsymbol{w}^\star$ is the minimum of $\ell_i \circ f_i$, $\mathbf{0}$ is the minimizer of $F$, hence since $\boldsymbol{d}(\gamma \ell_i, f_i)(\boldsymbol{w}^\star)$ is defined as the minimizer of $F$ it must be $\mathbf{0}$. For $\boldsymbol{d}(\gamma q_i^\star, f_i)(\boldsymbol{w}^\star)$, then $\nabla(\ell_i \circ f_i)(\boldsymbol{w}^\star) = \mathbf{0}$ so the direction reduces to compute $\boldsymbol{d}(\gamma q_i^\star, f_i)(\boldsymbol{w}^\star) = \operatorname{argmin}_{\boldsymbol{d} \in \mathbb{R}^p} \frac{1}{2} \langle \boldsymbol{d}, Q\boldsymbol{d} \rangle + \frac{1}{2\gamma} \|\boldsymbol{d}\|_2 = \mathbf{0}$ for $Q = \partial f_i(\boldsymbol{w}^\star)^* \nabla^2 \ell_i(f_i(\boldsymbol{w}^\star)) \partial f_i(\boldsymbol{w}^\star) \succeq 0$. On the other hand, we have for any $\boldsymbol{w}^\star$, $\boldsymbol{d}(\gamma q_i^\star, f_i)(\boldsymbol{w}^\star) = (\gamma^{-1} \operatorname{I} + Q)^{-1} \nabla(\ell_i \circ f_i)(\boldsymbol{w}^\star)$, so $\boldsymbol{d}(\gamma q_i^\star, f_i) = \mathbf{0} \iff \nabla(\ell_i \circ f_i)(\boldsymbol{w}^\star) = \mathbf{0}$. □

## C.6 Proof of Proposition 1 (descent direction, exact case)

We show a slightly stronger result, namely that, unless $\boldsymbol{d}(\gamma q_S^t, \boldsymbol{f}_S)(\boldsymbol{w}^t) = \mathbf{0}$, that is, $\boldsymbol{w}^t$ is a critical point of $\ell_S \circ \boldsymbol{f}_S$ as shown in Proposition 13, the direction $\boldsymbol{d}(\gamma q_S^t, \boldsymbol{f}_S)(\boldsymbol{w}^t)$ satisfies $\langle \boldsymbol{d}(\gamma q_S^t, \boldsymbol{f}_S)(\boldsymbol{w}^t), \nabla(\ell_S \circ \boldsymbol{f}_S)(\boldsymbol{w}^t) \rangle > 0$. The result claims in the main text holds naturally for $\boldsymbol{d}(\gamma q_S^t, \boldsymbol{f}_S)(\boldsymbol{w}^t) = \mathbf{0}$.

Denote

$$F_S^t(\boldsymbol{d}) := \ell_S(f_S(\boldsymbol{w}^t) - \partial f_S(\boldsymbol{w}^t)\boldsymbol{d}) + \frac{m}{2\gamma} \|\boldsymbol{d}\|_2^2$$

and $\boldsymbol{d}^\star := \boldsymbol{d}(\gamma \ell_S, f_S)(\boldsymbol{w}) = \operatorname{argmin}_{\boldsymbol{d} \in \mathbb{R}^p} F_S^t(\boldsymbol{d})$. Since $F_S^t$ is strongly convex and assuming $\boldsymbol{d}^\star \neq 0$, we have $F_S^t(\boldsymbol{d}^\star) > F_S^t(\mathbf{0}) + \nabla F_S^t(\mathbf{0})^\top \boldsymbol{d}^\star$, that is, $\nabla F_S^t(\mathbf{0})^\top \boldsymbol{d}^\star < F_S^t(\boldsymbol{d}^\star) - F_S^t(\mathbf{0}) < 0$ since $\boldsymbol{d}^\star = \operatorname{argmin}_{\boldsymbol{d} \in \mathbb{R}^p} F_S^t(\boldsymbol{d})$. Note that $\nabla F_S^t(\mathbf{0}) = -\partial f_S(\boldsymbol{w}^t)^* \nabla \ell_S(f_S(\boldsymbol{w}^t)) = -\nabla(\ell_S \circ f_S)(\boldsymbol{w}^t)$. Hence, $\langle \nabla(\ell_S \circ f_S)(\boldsymbol{w}^t), -\boldsymbol{d}^\star \rangle < 0$.

For the quadratic case, a similar reasoning applies. Denote now

$$G_S^t(\boldsymbol{d}) := q_S^t(\boldsymbol{f}_S^t - \partial f_S(\boldsymbol{w}^t)\boldsymbol{d}) + \frac{m}{2\gamma} \|\boldsymbol{d}\|_2^2.$$

We have that $G_S^t$ is strongly convex with minimizer $\boldsymbol{d}(\gamma q_S, \boldsymbol{f}_S)(\boldsymbol{w}^t)$. Moreover, we have that $\nabla G_S^t(\mathbf{0}) = -\partial f_S(\boldsymbol{w}^t)^* \nabla \ell_S(f_S(\boldsymbol{w}^t)) = -\nabla(\ell_S \circ f_S)(\boldsymbol{w}^t)$. Hence, as above, assuming $\boldsymbol{d}(\gamma q_S, \boldsymbol{f}_S)(\boldsymbol{w}^t) \neq 0$, we get that $\langle \nabla(\ell_S \circ f_S)(\boldsymbol{w}^t), -\boldsymbol{d}(\gamma q_S, f_S)(\boldsymbol{w}^t) \rangle < 0$.

## C.7 Proof of Proposition 2 (descent direction, inexact case)

**Primal case.**

**Proposition 14.** *Denote $\boldsymbol{d}_S^{t;\tau}$ the approximate solution of the following problem computed by a conjugate gradient method after $\tau$ iterations,*

$$\operatorname*{argmin}_{\boldsymbol{d}\in\mathbb{R}^p} \frac{1}{2}\langle\boldsymbol{d}, J^*HJ\boldsymbol{d}\rangle - \langle\boldsymbol{d}, J^*\boldsymbol{g}\rangle + \frac{m}{2\gamma}\|\boldsymbol{d}\|_2^2$$

*for $J := \partial\boldsymbol{f}_S(\boldsymbol{w}^t)$, $H := \nabla^2\ell_S(\boldsymbol{f}_i^t)$, $\boldsymbol{g} := \nabla\ell_S(\boldsymbol{f}_i^t)$, $\boldsymbol{f}_S^t := \boldsymbol{f}_S(\boldsymbol{w}^t)$, $\gamma > 0$, $m := |S|$. Then $-\boldsymbol{d}_S^{t;\tau}$ is a descent direction for $\ell_S \circ \boldsymbol{f}_S$ at $\boldsymbol{w}^t$.*

*Proof.* The problem at hand is a convex quadratic of the form $\min_{\boldsymbol{d}\in\mathbb{R}^p} \frac{1}{2}\langle\boldsymbol{d}, Q\boldsymbol{d}\rangle - \langle\boldsymbol{c}, \boldsymbol{d}\rangle$ for $Q := J^*HJ + (m/\gamma)\mathrm{I}$ and $\boldsymbol{c} := J^*\boldsymbol{g}$. Using Lemma 16, we have that the $\tau^{\text{th}}$ iterate of a conjugate gradient method applied to the above problem satisfies $\langle\boldsymbol{d}_S^{t;\tau}, \boldsymbol{c}\rangle \geq 0$. Since $\boldsymbol{c} = \nabla h_S(\boldsymbol{w}^t)$, the result follows. $\square$

**Dual case.**

**Proposition 15.** *Consider computing the prox-linear direction*

$$\boldsymbol{d}_S^t = \operatorname*{argmin}_{\boldsymbol{d}\in\mathbb{R}^p} \frac{1}{2}\langle\boldsymbol{d}, J^*HJ\boldsymbol{d}\rangle - \langle\boldsymbol{d}, J^*\boldsymbol{g}\rangle + \frac{m}{2\gamma}\|\boldsymbol{d}\|_2^2$$

*for $J = \partial\boldsymbol{f}_S(\boldsymbol{w}^t)$, $H = \nabla^2\ell_S(\boldsymbol{f}_i^t)$, $\boldsymbol{g} = \nabla\ell_S(\boldsymbol{f}_S^t)$, $\boldsymbol{f}_S^t = \boldsymbol{f}_S(\boldsymbol{w}^t)$, $\gamma > 0$, via its dual formulation as $\boldsymbol{d}_S^t = \frac{\gamma}{m}J^*\boldsymbol{\alpha}_S^t$, $\boldsymbol{\alpha}_S^t = \boldsymbol{g} - \boldsymbol{\beta}_S^t$ for*

$$\boldsymbol{\beta}_S^t = \operatorname*{argmin}_{\boldsymbol{\beta}\in\mathbb{R}^k} \frac{1}{2}\langle\boldsymbol{\beta}, P\left(H^\dagger + \frac{\gamma}{m}JJ^*\right)P\boldsymbol{\beta}\rangle - \frac{\gamma}{\beta}\langle\boldsymbol{\beta}, PJJ^*\boldsymbol{g}\rangle, \tag{17}$$

*and $P = HH^\dagger$ as presented in Corollary 11. Let $\boldsymbol{\beta}_S^{t;\tau}$ be the $\tau^{\text{th}}$ iteration of a conjugate gradient method applied to (17) with associated primal direction $\boldsymbol{d}_S^{t;\tau} = \frac{\gamma}{m}(J^*\boldsymbol{g} - J^*\boldsymbol{\beta}_S^{t;\tau})$. We have $\langle\boldsymbol{d}_S^{t;\tau}, \nabla(\ell_i \circ f_i)(\boldsymbol{w}^t)\rangle \geq 0$ so that $-\boldsymbol{d}_S^{t;\tau}$ is a descent direction for $\ell_S \circ \boldsymbol{f}_S$ at $\boldsymbol{w}^t$.*

*Proof.* For simplicity denote $\tilde{\gamma} = \gamma/m$. Consider problem (17) in a canonical form

$$\min_{\boldsymbol{\beta}\in\mathbb{R}^k} \frac{1}{2}\langle\boldsymbol{\beta}, Q\boldsymbol{\beta}\rangle - \langle\boldsymbol{\beta}, \boldsymbol{c}\rangle,$$

for $Q = P(H^\dagger + \tilde{\gamma}JJ^*)P$, $\boldsymbol{c} = \tilde{\gamma}PJJ^*\boldsymbol{g}$. Consider $\boldsymbol{\beta}^\tau$ the $\tau^{\text{th}}$ iteration of a conjugate gradient method applied to the above problem whose iterations are presented in Lemma 16. Our goal is to show that for any $\tau \geq 0$, $\langle\boldsymbol{d}^\tau, \nabla(\ell_S \circ \boldsymbol{f}_S)(\boldsymbol{w}^t)\rangle \geq 0$ for $\boldsymbol{d}^\tau = \tilde{\gamma}(J^*\boldsymbol{g} - J^*\boldsymbol{\beta}^\tau)$, which reads

$$\langle\boldsymbol{\beta}^\tau, JJ^*\boldsymbol{g}\rangle \leq \langle\boldsymbol{g}, JJ^*\boldsymbol{g}\rangle.$$

Note that given the forms of $Q$ and $\boldsymbol{c}$ above, the iterates of a conjugate gradient method satisfy $\boldsymbol{\beta}^\tau = P\boldsymbol{\beta}^\tau$ with $P$ an orthonormal projector satisfying $P = P^*$. The above condition is then equivalent to $\tilde{\gamma}^{-1}\langle\boldsymbol{\beta}^\tau, \boldsymbol{c}\rangle = \langle\boldsymbol{\beta}^\tau, PJJ^*\boldsymbol{g}\rangle \leq \langle\boldsymbol{g}, JJ^*\boldsymbol{g}\rangle$. We proceed by contradiction and assume there exists $\tau_0 \in \{0, 1, \ldots, +\infty\}$ such that

$$\langle\boldsymbol{\beta}^{\tau_0}, JJ^*\boldsymbol{g}\rangle > \langle\boldsymbol{g}, JJ^*\boldsymbol{g}\rangle. \tag{18}$$

Recall that, with the notations of Lemma 16, for any $\tau \geq 0$, $\tilde{\gamma}\langle\boldsymbol{p}^\tau, JJ^*\boldsymbol{g}\rangle = \langle\boldsymbol{p}^\tau, \boldsymbol{c}\rangle \geq 0$. If (18) is true, then, for any $\tau > \tau_0$, since $\boldsymbol{\beta}^\tau = \boldsymbol{\beta}^{\tau_0} + \sum_{s=\tau_0+1}^{\tau} a_s\boldsymbol{p}^{s-1}$, $a_s \geq 0$ and $\tilde{\gamma} \geq 0$, we have that $\langle\boldsymbol{\beta}^\tau, JJ^*\boldsymbol{g}\rangle > \langle\boldsymbol{g}, JJ^*\boldsymbol{g}\rangle$. Taking $\tau \to +\infty$, we have then

$$0 \leq \lim_{t\to+\infty}\langle J^*(\boldsymbol{\beta}^\tau - \boldsymbol{g}), J^*\boldsymbol{g}\rangle = \langle J^*(\boldsymbol{\beta}^\star - \boldsymbol{g}), J^*\boldsymbol{g}\rangle = \langle-\boldsymbol{d}^\star, J^*\boldsymbol{g}\rangle,$$

for $\boldsymbol{d}^\star$ the prox-linear direction. This contradicts Proposition 1 where in the proof we showed that the prox-linear direction satisfies $\langle\boldsymbol{d}^\star, J^*\boldsymbol{g}\rangle > 0$, unless $\boldsymbol{d}^\star = \boldsymbol{0}$, in which the case, the claim holds trivially. Hence, we have shown the claim, i.e., that for any $\tau \geq 0$, $\langle\boldsymbol{\beta}^\tau, JJ^*\boldsymbol{g}\rangle \leq \langle\boldsymbol{g}, JJ^*\boldsymbol{g}\rangle$. Therefore, the output primal direction $\boldsymbol{d}^\tau$ satisfies $\langle\boldsymbol{d}^\tau, \nabla(\ell_S \circ \boldsymbol{f}_S)(\boldsymbol{w}^t)\rangle \geq 0$. $\square$

**Lemma 16.** *Consider the iterations of a conjugate gradient method for solving $\min_{\boldsymbol{x} \in \mathbb{R}^d} \frac{1}{2}\langle \boldsymbol{x}, Q\boldsymbol{x} \rangle - \langle \boldsymbol{c}, \boldsymbol{x} \rangle$, i.e., starting from $\boldsymbol{x}^0 := \boldsymbol{0}$ and $\boldsymbol{r}^0 := \boldsymbol{p}^0 := \boldsymbol{c}$,*

$$
\begin{aligned}
a^\tau &:= \langle \boldsymbol{r}^{\tau-1}, \boldsymbol{r}^{\tau-1} \rangle / \langle \boldsymbol{p}^{\tau-1}, Q\boldsymbol{p}^{\tau-1} \rangle \\
\boldsymbol{x}^\tau &:= \boldsymbol{x}^{\tau-1} + a^\tau \boldsymbol{p}^{\tau-1} \\
\boldsymbol{r}^\tau &:= \boldsymbol{r}^{\tau-1} - a^\tau Q\boldsymbol{p}^{\tau-1} \\
b^\tau &:= \langle \boldsymbol{r}^\tau, \boldsymbol{r}^\tau \rangle / \langle \boldsymbol{r}^{\tau-1}, \boldsymbol{r}^{\tau-1} \rangle \\
\boldsymbol{p}^\tau &:= \boldsymbol{r}^\tau + b^\tau \boldsymbol{p}^{\tau-1}.
\end{aligned}
$$

*Then, for any $\tau \geq 0$, we have $\langle \boldsymbol{p}^\tau, \boldsymbol{c} \rangle \geq 0$ and*

$$\langle \boldsymbol{x}^\tau, \boldsymbol{c} \rangle \geq 0.$$

*Proof.* Since $\boldsymbol{x}^\tau = \boldsymbol{x}^0 + \sum_{s=1}^\tau a^s \boldsymbol{p}^{s-1}$ with $a_s \geq 0$ for all $s$ and $\boldsymbol{x}^0 = \boldsymbol{0}$, it suffices to show that $\langle \boldsymbol{p}^\tau, \boldsymbol{c} \rangle \geq 0$ for all $\tau \geq 0$. For $\tau = 0$, we have $\langle \boldsymbol{p}^0, \boldsymbol{c} \rangle = \|\boldsymbol{c}\|_2^2 \geq 0$. Assume $\langle \boldsymbol{p}^{\tau-1}, \boldsymbol{c} \rangle \geq 0$ for $\tau \geq 1$, then, as $b^\tau \geq 0$,

$$\langle \boldsymbol{p}^\tau, \boldsymbol{c} \rangle = \langle \boldsymbol{r}^\tau, \boldsymbol{c} \rangle + b^\tau \langle \boldsymbol{p}^{\tau-1}, \boldsymbol{c} \rangle \geq \langle \boldsymbol{r}^\tau, \boldsymbol{b} \rangle = \langle \boldsymbol{r}^\tau, \boldsymbol{r}^0 \rangle = 0,$$

where the last equality comes from the orthogonality of the residuals in a conjugate gradient method. The proof has then been shown by induction. $\qquad\square$

## D  Computational complexities

The computational complexities of the primal and dual formulation can be formally compared in the case of the squared loss or the quadratic approximation of the loss. We consider for simplicity here the formulation of the prox-linear step for squared losses, whose primal formulation reads, for a mini-batch $S = \{i_1, \ldots, i_m\}$,

$$
\boldsymbol{d}_S^t = \operatorname*{argmin}_{\boldsymbol{d} \in \mathbb{R}^p} \frac{1}{2} \langle \boldsymbol{d}, \left( (J_S^t)^* J_S^t + (m/\gamma)\, \mathrm{I} \right) \boldsymbol{d} \rangle - \langle (J_S^t)^* \boldsymbol{g}_S^t, \boldsymbol{d} \rangle, \tag{19}
$$

with associated dual formulation,

$$
\boldsymbol{\alpha}_S^t = \operatorname*{argmin}_{\boldsymbol{\alpha} \in \mathbb{R}^p} \frac{1}{2} \langle \boldsymbol{\alpha}, \left( J_S^t (J_S^t)^* + (\gamma/m)\, \mathrm{I} \right) \boldsymbol{\alpha} \rangle - \langle \boldsymbol{g}_S^t, \boldsymbol{\alpha} \rangle, \quad \boldsymbol{d}_S^t = (J_S^t)^* \boldsymbol{\alpha}_S^t, \tag{20}
$$

and associated shortcuts,

$$
\begin{aligned}
\boldsymbol{f}_S^t &= \boldsymbol{f}_S(\boldsymbol{w}^t) = (\boldsymbol{f}_{i_1}(\boldsymbol{w}^t), \ldots, \boldsymbol{f}_{i_m}(\boldsymbol{w}^t)) \in \mathbb{R}^{m \times k}, \\
\boldsymbol{g}_S^t &= \nabla \ell_S(\boldsymbol{f}_S^t) = (\nabla \ell_{i_1}(\boldsymbol{f}_1^t), \ldots, \nabla \ell_{i_m}(\boldsymbol{f}_m^t)) \in \mathbb{R}^{m \times k}, \\
J_S^t \boldsymbol{u} &= \partial \boldsymbol{f}_S(\boldsymbol{w}^t)(\boldsymbol{u}) = (J_{i_1}^t \boldsymbol{u}, \ldots, J_{i_m}^t \boldsymbol{u}), \quad J_i^t \boldsymbol{u} = \partial \boldsymbol{f}_i(\boldsymbol{w}^t)\boldsymbol{u}, \quad \text{for } \boldsymbol{u} \in \mathbb{R}^p, \\
(J_S^t)^* \boldsymbol{v} &= (\partial \boldsymbol{f}_S(\boldsymbol{w}^t))^* \boldsymbol{v} = \sum_{j=1}^m (J_{i_j}^t)^* \boldsymbol{v}_j, \quad \text{for } \boldsymbol{v} = (\boldsymbol{v}_1, \ldots, \boldsymbol{v}_m) \in \mathbb{R}^{m \times k}.
\end{aligned}
$$

For each formulation, we need to consider (i) the cost of each iteration of the inner solver, (ii) the condition number of each subproblem.

**Computational cost of running $\tau$ iterations of the inner solver.** Consider a conjugate gradient (CG) applied to solve $\min_{\boldsymbol{x} \in \mathbb{R}^d} \frac{1}{2}\langle \boldsymbol{x}, Q\boldsymbol{x} \rangle - \langle \boldsymbol{x}, \boldsymbol{c} \rangle$ with $Q$ positive definite as recalled in Lemma 16. Each iteration of CG requires

1. 1 call to the linear operator $\boldsymbol{x} \mapsto Q\boldsymbol{x}$ to compute $Q\boldsymbol{p}^{\tau-1}$; if this linear call amounts to a matrix-vector computation, it costs $O(d^2)$ elementary computations,

2. 3 inner products computations in $\mathbb{R}^d$, each at a cost of $O(d)$ elementary computations, to compute $\langle \boldsymbol{r}^{\tau-1}, \boldsymbol{r}^{\tau-1} \rangle, \langle \boldsymbol{r}^\tau, \boldsymbol{r}^\tau \rangle, \langle \boldsymbol{p}^{\tau-1}, Q\boldsymbol{p}^{\tau-1} \rangle,$

3. 3 additions in $\mathbb{R}^d$, each at a cost of $O(d)$ elementary computations, to compute $\boldsymbol{x}^\tau, \boldsymbol{r}^\tau, \boldsymbol{p}^\tau$.

For the primal formulation (19), computing the linear operator call $\boldsymbol{x} \mapsto Q_{\text{primal}}\boldsymbol{x}$, with $Q_{\text{primal}} \coloneqq (J_S^t)^* J_S^t + (m/\gamma)\,\mathrm{I}$ amounts to one call to the JVP $J_S^t$ and one call to the VJP $(J_S^t)^*$. For the dual formulation, computing the linear operator call $\boldsymbol{x} \mapsto Q_{\text{dual}}\boldsymbol{x}$, with $Q_{\text{dual}} \coloneqq J_S^t(J_S^t)^* + (\gamma/m)\,\mathrm{I}$ amounts to one call to the VJP $(J_S^t)^*$ and one call to the JVP $J_S^t$. Hence, the computation of the linear operator is the same in both formulations.

Other computational costs differ whether the primal or the dual formulation is considered. For the primal formulation (19), the variables are of the dimension of the parameters, that is all other computations incur a cost of $O(p)$ elementary computations. On the other hand, for the dual formulation (20), the variables are of dimension the number of samples times the output dimension, that is all other computations incur a cost of $O(mk)$ in this case. For small batches, we retrieve that the dual formulation can be advantageous.

Finally, each formulation requires one additional call to the VJP $(J_S^t)^*$. For the primal formulation, this call is necessary to compute $(J_S^t)^* \boldsymbol{g}_S^t$. For the dual formulation, this call is necessary to map back the dual solution to the primal solution, i.e., computing $(J_S^t)^* \boldsymbol{\alpha}_S^t$.

To summarize, the computational cost of running $\tau$ iterations of CG to compute the prox-linear update is

$$\text{Primal formulation: } \tau(\mathcal{T}(\partial \boldsymbol{f}_S(\boldsymbol{w}^t)) + \mathcal{T}(\partial \boldsymbol{f}_S(\boldsymbol{w}^t)^*) + O(p)) + \mathcal{T}(\partial \boldsymbol{f}_S(\boldsymbol{w}^t)^*),$$
$$\text{Dual formulation: } \tau(\mathcal{T}(\partial \boldsymbol{f}_S(\boldsymbol{w}^t)) + \mathcal{T}(\partial \boldsymbol{f}_S(\boldsymbol{w}^t)^*) + O(mk)) + \mathcal{T}(\partial \boldsymbol{f}_S(\boldsymbol{w}^t)^*),$$

where $\mathcal{T}(\partial \boldsymbol{f}_S(\boldsymbol{w}^t))$ and $\mathcal{T}(\partial \boldsymbol{f}_S(\boldsymbol{w}^t)^*)$ denote the computational complexity of the JVP and VJP of $f_S$ respectively in a differentiable programming framework. This computational complexity varies with the architecture considered. As an illustrative example, if the network considered is a fully connected network with $D$ layers of constant input-output dimensions $H$, the computational cost of a JVP/VJP is of the order of $O(DH^2)$ elementary computations.

**Conditioning of primal and dual formulations** The convergence rate of a CG method on a problem of the form $\min_{\boldsymbol{x} \in \mathbb{R}^d} \frac{1}{2}\langle \boldsymbol{x}, Q\boldsymbol{x} \rangle - \langle \boldsymbol{x}, \boldsymbol{c} \rangle$ with $Q$ positive definite is theoretically given by

$$\|\boldsymbol{x}^\tau - \boldsymbol{x}^*\| \leq 2\left(\frac{\sqrt{\kappa}-1}{\sqrt{\kappa}+1}\right)^\tau \|\boldsymbol{x}^0 - \boldsymbol{x}^*\|_2, \quad \text{for } \kappa = \frac{\lambda_{\max}(Q)}{\lambda_{\min}(Q)}$$

where $\boldsymbol{x}^* = \operatorname{argmin}_{\boldsymbol{x} \in \mathbb{R}^d} \frac{1}{2}\langle \boldsymbol{x}, Q\boldsymbol{x} \rangle - \langle \boldsymbol{x}, \boldsymbol{c} \rangle$, $\kappa$ is the condition number associated to the subproblem, and $\lambda_{\max}(Q), \lambda_{\min}(Q)$ are respectively the largest and smallest eigenvalues of $Q$.

In our case, $\lambda_{\max} \coloneqq \lambda_{\max}(J_S^t(J_S^t)^*) = \lambda_{\max}((J_S^t)^* J_S^t)$. If $p > m \times k$, we have that necessarily $\lambda_{\min}((J_S^t)^* J_S^t) = 0$, while $\lambda_{\min} \coloneqq \lambda_{\min}(J_S^t(J_S^t)^*) \geq 0$. Hence, the condition numbers associated to the primal (19) and dual (20) formulations are respectively, in the case where $p > m \times k$,

$$\text{Primal: } \kappa_{\text{primal}} = 1 + \frac{\gamma \lambda_{\max}}{m} \qquad \text{Dual: } \kappa_{\text{dual}} = \frac{\lambda_{\max} + (\gamma/m)}{\lambda_{\min} + (\gamma/m)}.$$

The dual condition number can thus take advantage of the minimal eigenvalue $\lambda_{\min}$ of $J_S^t(J_S^t)^*$. In practice, the rate of CG depends highly on the distribution of eigenvalues of the linear operator considered (Strakovs, 1991; Greenbaum, 1997). Typically, if the eigenvalues are clustered rather than dispersed, the CG method can converge must faster (Greenbaum, 1997).

## E Experimental Details

**Architecture.** The ConvNet we consider in Section 5 is defined by (i) a convolutional layer with 32 filters of kernel size $3 \times 3$ followed by the SiLU activation function (Elfwing et al., 2018) and an average pooling layer of kernel size $2 \times 2$, (ii) another convolutional layer with 64 filters of kernel size $3 \times 3$ followed by the SiLU and an average pooling layer of kernel size $2 \times 2$, (iii) a dense layer of output dimension 256, followed by the SiLU, (iv) a final dense layer for classification of output size 10.

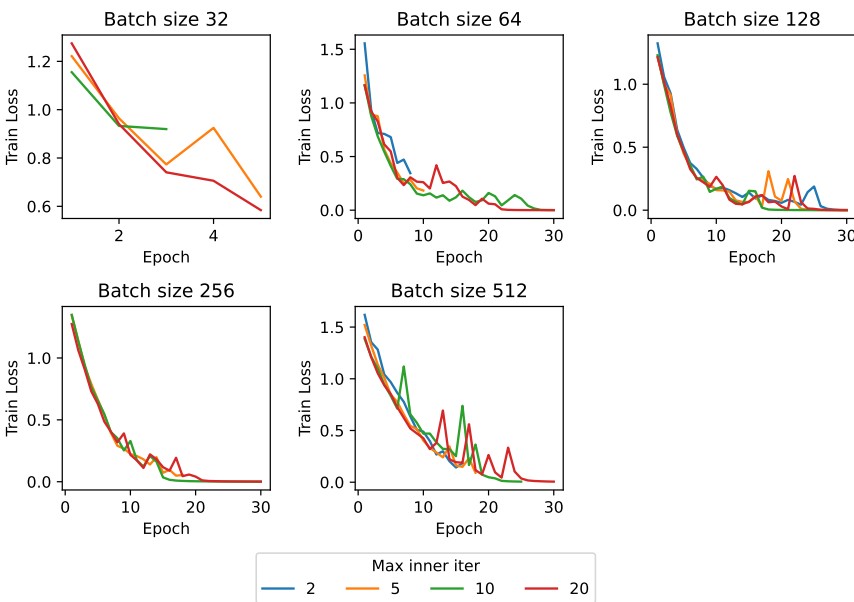

Figure 6: Sensitivity to inner iterations across batch-sizes for SPL without linesearch.

**Computing infrastructure** The experiments have been run on TPUv2 (180 TFLOPS) and TPUv3 (420 TFLOPS) machines with eight tensor nodes that is a single tray.

## F Additional experiments

### F.1 Diagnosis experiments

**Sensitivity to inner iterations across batch-sizes.** Figure 6 displays the performance of the **SPL** algorithm using descent directions for a fixed stepsize $\gamma = 1$ and varying numbers of inner iterations and batch-sizes. We observe that for small mini-batches or very large mini-batches (at 1024, none of the choice of inner iterations led to convergence), the algorithm suffers from numerical instabilities as it stops suddenly. Adding an Armijo-type line-search stabilizes the algorithm across the board. Overall, a medium batch-size (here 128 or 256) appears best performing.

**Plots in time.** Figure 7 presents the results of Figure 1 in time for completeness.

### F.2 Additional architectures

We consider here several other architectures ranging from CNNs with various depths and a larger ResNet model.

**ResNets on ImageNet.** We consider the ResNet 18 and 34 architecture as presented by He et al. (2016), except that we consider SiLU activation functions (Elfwing et al., 2018) instead of the ReLU activation function. We consider standard preprocessing of images from ImageNet (crop and center images randomly to a $224 \times 224$ size).

To test the performance of the prox-linear, we consider its implementation (i) with various $\gamma$ ranging in $\{10^i, i \in \{-2, \ldots, 2\}\}$, (ii) with 1 or 2 inner iterations, (iii) with (**Armijo SPL**) or without additional linesearch (**SPL**). We consider 2 inner iterations for the inner solver. We test SPL against SGD, SGD with momentum and Adam, whose learning rates are searched on a log 10 scale in $\{10^i, i \in \{-5, \ldots, 0\}\}$.

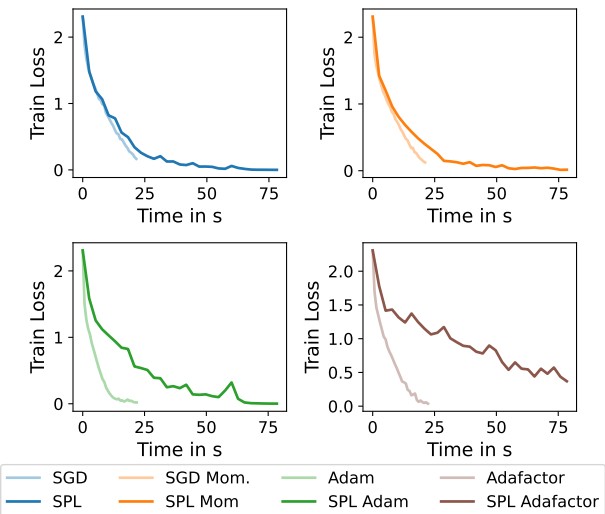

Figure 7: Prox-linear direction as a replacement for the stochastic gradient in existing algorithms, in time.

The results are presented on Figure 8. We observe that the potential gains offered by a Gauss-Newton method such as SPL are not apparent in this architecture, though SPL still optimizes well the objective.

**CNNs on CIFAR10 with various depths.** We consider additional experiments on the CIFAR10 datasets with convolutional networks of various depths consisting in $x$ layers of 128 filters, $x$ layers of 64 filters, $x$ layers of 32 filters and a final dense layer, for $x \in \{1, 2, 3\}$, corresponding to "short", "medium", "long" architectures respectively. Each layer is followed by a SiLU activation function (Elfwing et al., 2018). All convolutional layers have a $3 \times 3$ kernel.

In each setting, we consider an implementation of SPL (i) with various $\gamma$ ranging in $\{10^i, i \in \{-3, \ldots, 3\}\}$, (ii) with 1, 2, 4 or 6 inner iterations, (iii) with (Armijo SPL) or without additional linesearch (SPL). We test SPL against SGD, SGD with momentum, Adam, Shampoo and KFAC, whose learning rates are searched on a log 10 scale in $\{10^i, i \in \{-5, \ldots, 0\}\}$. For KFAC, we consider a fixed momentum of 0.9, a fixed damping of $10^{-3}$ and used the implementation available at `https://github.com/google-deepmind/kfac-jax` with support for parallelism on accelerators.

The results are presented on Figures 9, 10, 11. We observe that SPL-Armijo performs generally on par with its competitors, performing best for small bath-sizes. For small batch-sizes, we also observed that the best hyperparameters were generally to fix $\gamma$ to 1., and the number of inner iterations to 2. On the other hand, for larger batch-sizes, we observed that the performance of Armijo-SGD can deteriorate. Interestingly, we observed that for larger models and batch-sizes the best performances were obtained for larger number of inner iterations and smaller $\gamma$.

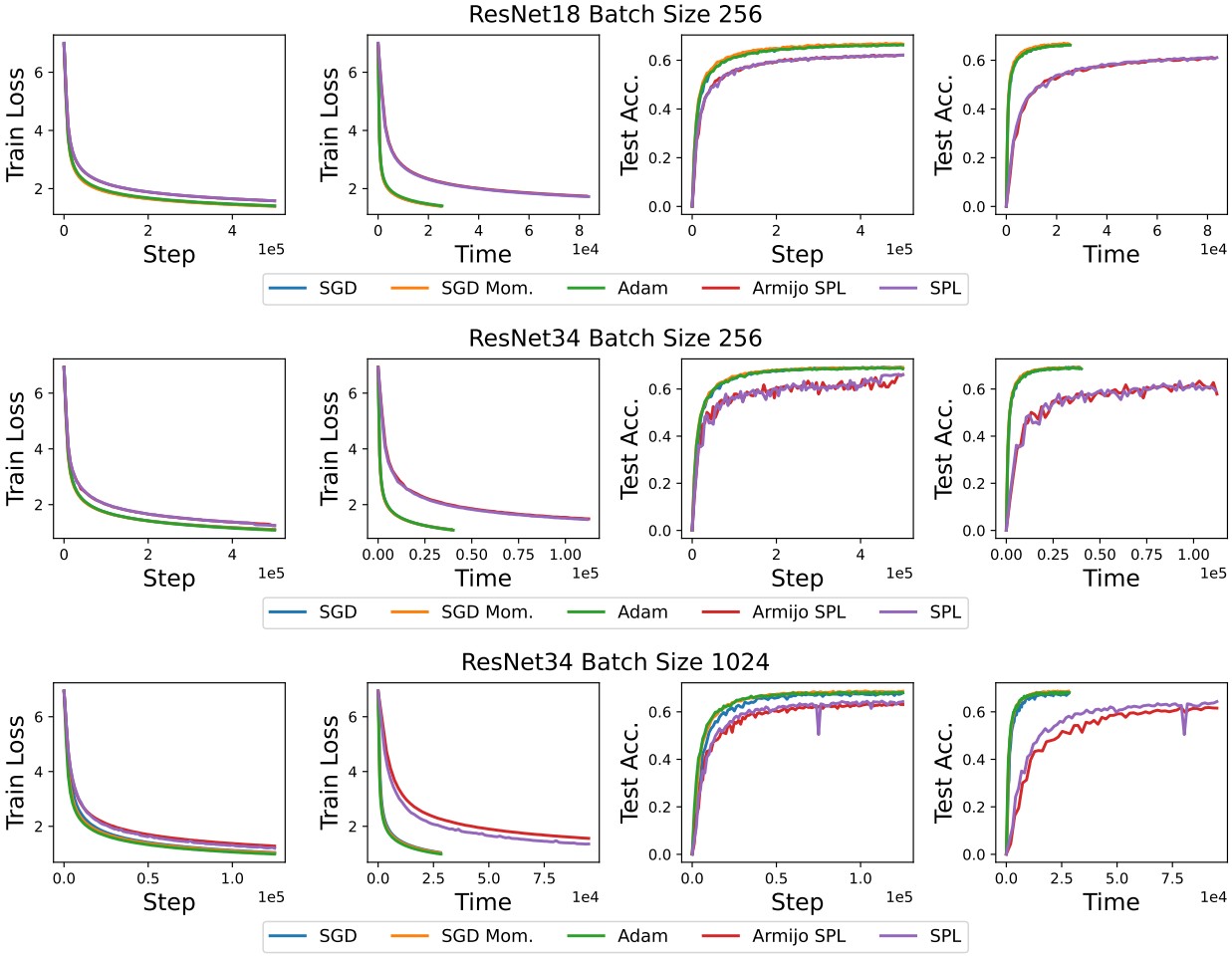

Figure 8: ImageNet. Top: ResNet18 batch-size 256, Middle: ResNet34 batch-size 256, Bottom: ResNet34 batch-size 1024, for various regularizations $\gamma$, various maximum inner iterations $\tau$.

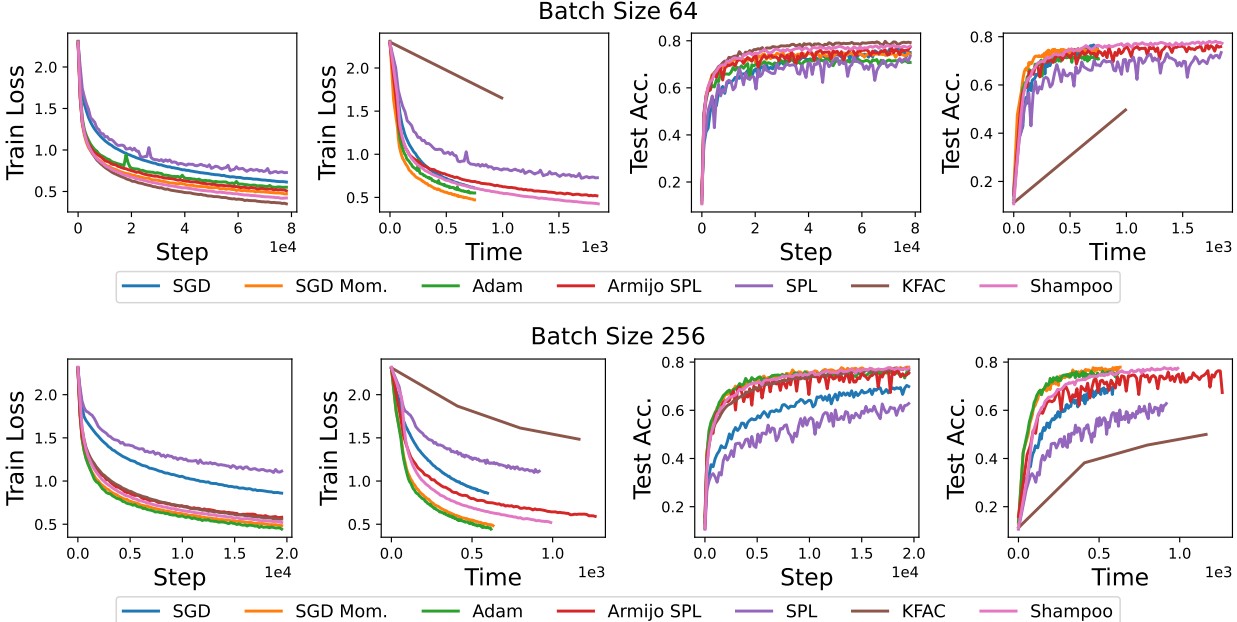

Figure 9: Short CNN $(128 \times 1, 64 \times 1, 32 \times 1)$ Top: Batch-size 64, Middle: Batch-size 256, Bottom: Batch-size 1024. Best of various regularizations $\gamma$ in $\{10^i, i \in \{-3, \dots, 3\}\}$. Best of various max inner iterations $\tau$ in $(1, 2, 4, 6)$.

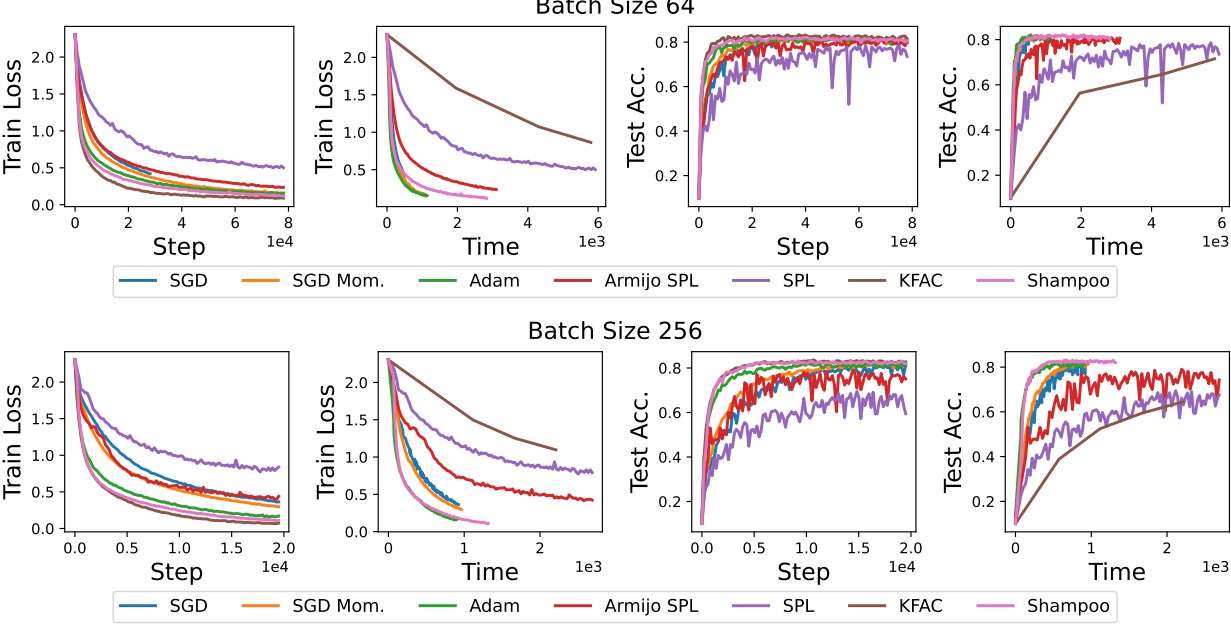

Figure 10: Medium CNN $(128 \times 2, 64 \times 2, 32 \times 2)$ Top: Batch-size 64, Middle: Batch-size 256, Bottom: Batch-size 1024. Best of various regularizations $\gamma$ in $\{10^i, i \in \{-3, \dots, 3\}\}$. Best of various max inner iterations $\tau$ in $(1, 2, 4, 6)$.

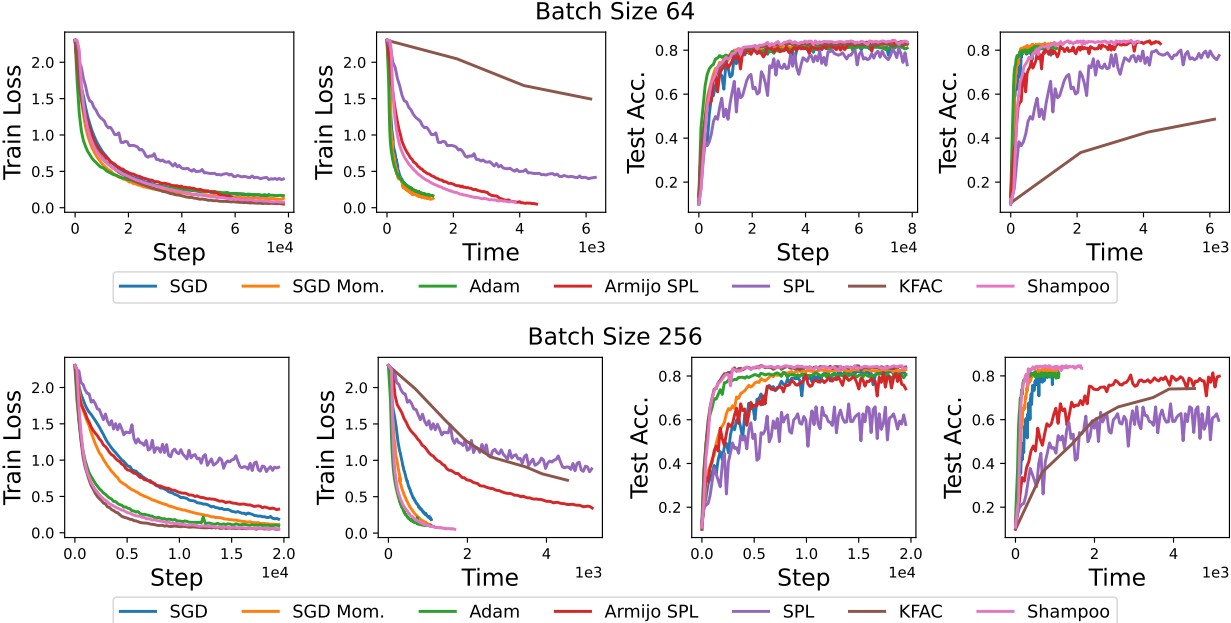

Figure 11: Long CNN $(128 \times 3, 64 \times 3, 32 \times 3)$. Top: Batch-size 64, Middle: Batch-size 256, Bottom: Batch-size 1024. Best of various regularizations $\gamma$ in $\{10^i, i \in \{-3, \ldots, 3\}\}$. Best of various max inner iterations $\tau$ in $(1, 2, 4, 6)$.

