# OpenReview forum: "Dual Gauss-Newton Directions for Deep Learning"
_TMLR — Rejected by TMLR_

### Review · Reviewer_C3Q1 · 2024-08-01

**Summary Of Contributions:**

This manuscript looks at decomposing the typical deep learning objective into the composition of two functions, and solving optimization steps more explicitly and with less approximation error.  The key idea is that we often apply a relatively simple function to estimate the loss (such as a BCE), and we can maintain the non-linearity of this function to help find a better descent direction.  This creates an optimization subproblem that can be solved efficiently through a dual formulation.  Empirical results as of yet do not show clear improvements, but may provide insights into future directions.

**Audience:**

Yes

**Claims And Evidence:**

No

**Requested Changes:**

I suggest the authors add:
- Details on when the descent direction can greatly change and really be better
- Convergence theory on the method overall
- More empirical results where methods have been run to convergence.

**Strengths And Weaknesses:**

This manuscript is fairly clear and the idea is communicated well.  The update steps are well-thought out and can be efficiently applied.

However, there are a number of issues that hinder the utility of the manuscript:

- It is unclear when the descent direction will be greatly changed through this optimization procedure.  I would encourage the authors to explain when this will make a big impact and greatly change optimization updates.  For example, this doesn't seem like it would change the direction of the squared error loss, and updates on something like a sigmoid function do not seem like it would greatly change the updates.
- In Section 4, the worst case results are pretty different, but I do not feel that the worst case really reflects the reality of improvement.  I would be interested in how this translates to real empirical cases--do we see such large differences in practice?

- It should be discussed how this algorithm can change convergence rates of a problem overall.  e.g., how much of an improvement can this provide?

-The empirical evidence is fairly light, and does not seem to support the promising nature of the algorithm in practice.  For example, in Figure 5, the algorithms are not run to convergence, so the claim that SPL leads to better test performance appear specious.  This is also fairly poor performance on this task in general.   The supplemental figures suggest that test performance does not improve from this method on more complex networks, not does it appear to be time competitive.

---

### Review · Reviewer_fCTm · 2024-09-03

**Summary Of Contributions:**

This paper considers the task of computing the prox-linear directions in deep learning, including the directions obtained via the so-called convex linear and quadratic-linear approximations. The paper proposes computing the prox-linear directions through their dual formulations. In this way, the dimension of the corresponding optimization problems is changed from $p$, the number of network parameters, to $m k$, the mini-batch size multiplied by the number of network outputs. This adjustment can be computationally beneficial when mk is much smaller than p, which is typically the case.

**Audience:**

Yes

**Broader Impact Concerns:**

This paper introduces a trick for neural network training, and I believe there are no broader impact concerns.

**Claims And Evidence:**

No

**Requested Changes:**

Please address the weaknesses above.

**Strengths And Weaknesses:**

**Strengths.**
1. The idea is simple yet the benefit is obvious.

----

**Weaknesses.**

1. While a simple yet effective solution is valued, I am not sure whether this is nearly trivial to related experts. The authors might need to highlight some technical challenges.
2. The paper is written sloppily. Below are some examples:
    - The term “Fenchel-Rockafellar dual” is mentioned only in the first and last sections, and does not appear in the main text, forcing the reader to guess what it refers to.
    - In the statement of contributions in Section 1, the authors mention that they proved “the proposed algorithm produces a descent direction when run for any number of steps.” It is not until Proposition 2 in Section 4 that the reader understands what this sentence means.
    - There are no citations or pointers to proofs for the dual formulations in Sections 3.1 and 3.2.
    - There is no citation or pointer to a proof for Proposition 1.
    - The proof of Proposition 2 is incorrectly referred to Appendix C.6. It should be Appendix C.7.
    - It is unclear why the discussion in Section C.2 is restricted to a specific strongly convex regularizer (the so-called “our case”).
    - The proof of Proposition 14 uses Lemma 16, where appears much later, without any reason.
3. The connection between the main text and the appendix needs to be stronger. The main text does not provide adequate guidance for the reader to navigate the appendix.
3. In the proof of Proposition 3, the validity of exchanging min and max should be justified; it is unclear where strong convexity is necessary.
4. Typo: A comma is missing in the third line of the first equation in Section C.3.

---

### Review · Reviewer_1ZgN · 2024-09-28

**Summary Of Contributions:**

The authors study the problem of using partial-linearization of loss objectives in order to obtain new descent directions for use in iterative optimization algorithms for training neural networks. They position their work within a broader sequence of papers examining related methods.

When training a neural network, the objective being minimized is a composition of a loss function with the neural network forward map. The authors approximate this objective by composing a quadratic approximation of the loss function with a linear approximation to the neural network's forward map to define a convex quadratic optimization subproblem. The hope is that at any iteration, solutions to this subproblem would provide better descent directions than the gradient of the original objective. This approach of approximating the objective with such a convex subproblem has roots in traditional Gauss-Newton algorithms. It is not clear to me whether the authors are the first to study the specific convex quadratic subproblem formulated in their paper (Eq. 4), or whether other works have studied this subproblem as well (particularly the the work cited by the authors, ``On the Promise of the Stochastic Generalized Gauss-Newton Method for Training DNNs''). In specific settings, the (dual) subproblem reduces to subproblems that arise in other iterative optimization algorithms (stochastic dual coordinate ascent), a potentially interesting connection. **I would appreciate if the authors could clarify to me whether they believe the subproblem formulation in Eq. 4 is itself is a novel contribution.**

In practice, this subproblem itself must be solved iteratively in order to produce a proposed update direction for the neural network. The authors show that in the setting where all loss functions are convex, the direction produced at any iteration of this solving procedure will be a descent direction. This analytical result reduces concerns about needing to solve each subproblem very exactly, and instead opens up users to trading off between solving the subproblem very precisely (at higher computational expense), and taking only a few iterative steps to solve the subproblem (and yielding a descent direction that does not minimize the subproblem as well).

Given this subproblem, one core novel contribution by the paper is to take advantage of its convexity to formulate the dual to the subproblem. A key advantage of considering the dual is that the number of variables involved in the dual subproblem is equal to the number of network outputs times the number of input samples. In contrast, the number of variables involved in the primal problem is the number of network parameters, which is typically much larger. This implies that if solving the convex subproblem in order to obtain a descent direction, solving the dual instead of the primal may be faster. The authors give a precise accounting for the computational complexities of solving the primal and dual problems which highlights this distinction, and the authors include some experimental evidence for this speedup in practice.

In order to efficiently solve the dual subproblem, the authors propose to formulate the dual as an equality-constrained quadratic program, to which they can apply conjugate gradient descent.

They also note that their method for producing a descent direction can be coupled with linesearch to introduce and set an additional stepsize parameter. This method is referred to as Armijo-SPL. Their base method, which uses the update produced by solving the convex subproblem without any additional stepsize parameter is referred to as SPL (stochastic prox-linear).

The authors present a suite of experiments examining the use of their method in training a convolutional neural network on the Cifar10 dataset. These experiments aim to probe several questions:
- Does the descent direction produced by their method out-perform the traditional stochastic gradient?
- If their descent direction is coupled to other algorithms (e.g. adding momentum, or using in place of the traditional gradient when running Adam and Adafactor) does it improve performance compared to using the traditional stochastic gradient?
- What are the sensitivities of their method to the number of inner iterations used to solve the subproblem? Or to the size of the batches used in training? Or to step size? For this last question, they consider multiplying the descent direction produced by solving the subproblem by some constant step size.
- How does the runtime of their method compare with other methods?

**Audience:**

Yes

**Broader Impact Concerns:**

I have no broader impact concerns for this work.

**Claims And Evidence:**

Yes

**Requested Changes:**

**Critical:** Some basic formatting issues must be addressed. Captions of Fig 1. and Fig. 2 are mixed together and unreadable. Layout of Figure 3 should be re-arranged to be consistent with caption or caption should be changed [(left) and (right) instead of (top) (bottom)].

**Strengthen:** One main contribution of the work is the consideration of the dual subproblem, and the computational advantage in solving the dual over solving the primal. One way the authors could emphasize this contribution is with a figure, similar to Figure 4, examining how the gap between the runtime of training the primal and dual solvers widens as the number of parameters in the network grows.

I also have several clarifying questions about the details of different experimental figures. I would appreciate if the authors would be willing to clarify these points.
- Apart from Fig. 4, should the reader assume that all SPL-based methods are solving the dual to produce descent directions?
- Figure 1: What was the impact on test accuracy? I see figures examining test accuracy for other experiments in the appendix, but none seem to be the counterpart of Figure 1.
- Figure 2. For both SPL primal and dual, the best-performing value for inner stepsize also appears to be the largest value tested. Is there continued improvement with even larger values?
- Figure 4. Presents train loss versus time in s., and purports to show that the SPL dual saves some run time. What was the termination criteria for these figures? Were both methods run for the same total number of iterations?
- Figure 3
(left). What was the termination criterion used for each trial? It does not appear that all trials for varying values of max inner iteration ran for the same number of epochs. I raise this because in the corresponding loss-versus-time figure to the right, the traces appear in a very attractive sequence of increasing runtime by max-inner-iterations, but if all methods are running for a different number of total iterations then it is difficult to interpret this trend.
- Fig. 3 (right). The authors present a hypothesis for the relationship between batch size and performance of the prox-linear direction. They conjecture that with large batches, a few steps of the subroutine may not be sufficient to get a good direction: can they defend this hypothesis by showing that with more steps this sensitivity to large batch sizes is reduced? In addition, I also have a similar question as on Fig. 3 (left) about termination criteria; the largest batch size appears to have had the shortest runtime?
- Figure 5. The authors claim the SPL-based methods can reach higher test accuracy, but it is difficult to interpret this figure without understanding what was the termination condition for each method. Were all run for the same number of epochs? This would be a more compelling figure if the authors can show that if the other methods are allowed to run for the same total time as the SPL-based methods, they still will not reach comparable levels of test accuracy.

**Strengths And Weaknesses:**

**Strengths:** It is my impression that the authors are the first to consider solving the dual subproblem in order to produce a descent direction. Given existing interest in Gauss-Newton type approaches for training neural networks, the formulation of this dual subproblem is interesting as the computational complexity scales much more conservatively with the size of the neural network, which might enable future researchers to consider prox-linear descent directions in otherwise prohibitively large architectures. More broadly I believe the work gives an interesting example of how convex duality can be leveraged in the analysis and design of algorithms for training neural networks, in which duality does not often arise due to lack of convexity.

**Weaknesses:** While I find the dual formulation interesting, I believe the greatest weakness of the paper is that neither the analysis nor the experimental results present strongly compelling evidence that prox-linear directions offer an advantage over traditional gradients, either on their own or incorporated into existing algorithms. For example, in my opinion the figure with the most compelling evidence in support of the use of prox-linear directions is Fig. 1, which shows that using prox-linear directions enables faster minimization of training loss over epochs than traditional stochastic gradients, and performs comparably to traditional gradients when used in Adam. However, the experiments also highlight that using such prox-linear directions opens the door to new downsides, including sensitivity to batch size, and more expensive runtime.

I consider this work to contain interesting ideas, so the limited support for the method provided by the experiments is not prohibitive to me. However, I do think that some basic steps could be taken to strengthen the experimental results and/or make them more transparent. Increased transparency in the experimental methodology would also strengthen the paper, as it would make it easier to evaluate how compelling the experimental evidence is.

---

### Public Comment · ~Alex_Shtoff1 · 2025-02-02
**Prior work**

It is a little unfortunate, but it appears that a prior work has already studies solving the proximal step of this form via its dual. See Section 4 of this Mathematical Programming Computation paper by Shtoff (2024): https://link.springer.com/article/10.1007/s12532-024-00258-8

It's also available on ArXiv link: https://arxiv.org/abs/2205.01457

The current paper under review appears to provide an approximate-dual derivation, which appears interesting, and conducts experiments on practical ML problems demonstrating the effectiveness of the approach, but the basic idea of solving the proximal step of the form in this paper via the dual, and the insights it brings, is not new.

Please consider it when reviewing and making recommendations.

---

> ### Author Response · Authors · 2025-02-02
> **Overlap with pointed out paper is not that large**
>
> We already had a discussion with you on social media about this and you seemed to agree.
>
> Your paper focuses on deriving the prox of $ h(\langle a, x \rangle + b)$ in the dual. You get closed forms because $a$ is a vector. In the prox-linear setting of our paper, this would mean that you only support scalar-valued networks.
>
> In our case, we need the prox of $h(Ax + b)$, where $A$ is the Jacobian of the network, seen as a linear map. There is no closed form in this case...
>
> Apologies for not citing your work though. We are going to fix this.

---

> > ### Public Comment · ~Alex_Shtoff1 · 2025-02-02
> > **Reply**
> >
> > Indeed I missed an important point - in your paper you indeed discuss $h(z)$ for a _vector_ $z$.

---

### Decision · Action_Editor_a4SF · 2025-01-22

**Recommendation:** Reject

**Comment:**

The paper proposes a conceptually simple dual idea to speed up the computation of prox-linear directions with applications to the training of neural networks. The reviewers appreciated the theoretical ideas in the paper. A number of clarifications and a more detailed statement of the limitations of the methods are necessary. Alternatively, more evidence for the benefits of the proposed method would be required. Please see the questions and requested changes put forward by the reviewers.

**Audience:**

The paper will be of interest to a subset of TMLR's audience.

**Claims And Evidence:**

The claims made are partially supported. More experiments and a clearer statements of the limitations of the proposed approach is needed  for the experimental analysis.

**Resubmission Of Major Revision:**

The authors may consider submitting a major revision at a later time.